# OmniSep: Unified Omni-Modality Sound Separation with Query-Mixup

**Xize Cheng**[1]  **Siqi Zheng**[2]  **Zehan Wang**[1]  **Minghui Fang**[1]  **Ziang Zhang**[1]
**Rongjie Huang**[1]  **Shengpeng Ji**[1]  **Jialong Zuo**[1]  **Tao Jin**[1]  **Zhou Zhao**[1*]
[1]Zhejiang University    [2]Alibaba Group
chengxize@zju.edu.cn    ck@mail.harvard.edu

## Abstract

Query-based sound separation (QSS) effectively isolate sound signals that match the content of a given query, enhancing the understanding of audio data. However, most existing QSS methods rely on a single modality for separation, lacking the ability to fully leverage homologous but heterogeneous information across multiple modalities for the same sound signal. To address this limitation, we introduce **Omni**-modal Sound **Sep**aration (**OmniSep**), a novel framework capable of isolating clean soundtracks based on omni-modal queries, encompassing both single-modal and multi-modal composed queries. Specifically, we introduce the `Query-Mixup` strategy, which blends query features from different modalities during training. This enables OmniSep to optimize multiple modalities concurrently, effectively bringing all modalities under a unified framework for sound separation. We further enhance this flexibility by allowing queries to influence sound separation positively or negatively, facilitating the retention or removal of specific sounds as desired. Finally, OmniSep employs a retrieval-augmented approach known as `Query-Aug`, which enables open-vocabulary sound separation. Experimental evaluations on MUSIC, VGGSOUND-CLEAN+, and MUSIC-CLEAN+ datasets demonstrate effectiveness of OmniSep, achieving state-of-the-art performance in text-, image-, and audio-queried sound separation tasks. For samples and further information, please visit the demo page at `https://omnisep.github.io/`.

## 1 Introduction

The development of sound separation (Kavalerov et al., 2019; Wisdom et al., 2020) has significantly advanced the ability to isolate and analyze specific sound signals from complex audio mixtures. Early sound separation methods focused on isolating specific audio signals like vocals, drums, and bass (Défossez et al., 2019; Zeghidour & Grangier, 2021; Wang et al., 2023b). Later, by employing a variety of queries to guide sound separation, Query-Based Sound Separation (QSS) effectively isolates sound signals that align with the given query, thereby enhancing the semantic understanding and interpretability of audio. Researchers delved into using natural language queries to extract semantically consistent audio tracks from audio sources, known as Text-Queried Sound Separation (TQSS) (Ochiai et al., 2020; Kong et al., 2020; Liu et al., 2022). However, the limitations of textual descriptions in conveying nuanced scene information spurred further exploration into leveraging visual content as queries for extracting object-emitted sounds from images, termed Image-Queried Sound Separation (IQSS) (Tzinis et al., 2020; Dong et al., 2022; Chen et al., 2023). Additionally, some researchers also attempted to employ audio references as queries for extracting similar audio tracks, denoted as Audio-Queried Sound Separation (AQSS) (Lee et al., 2019; Chen et al., 2022).

Despite these advances, we still face several challenges to scale up audio data with sound separation: **(1) Lack of a unified model to handle composed queries from multiple modalities.** Current sound separation methods rely solely on single-modal queries, limiting their effectiveness in accurately expressing the target sound signal. To overcome this limitation, we need a model that can accommodate multi-modal composed queries, thereby improving its ability to capture and express the nuances of the desired sound signal. **(2) Limited sound manipulation flexibility.** Current methods are limited

---

[*]Corresponding author.

to preserving specific sound signals based on queries but are unable to filter out specific sounds based on information corresponding to undesired sounds. **(3) Incapable of handling open-vocabulary queries.** Most of the existing audio-text datasets (Chen et al., 2020; Gemmeke et al., 2017) only offer predefined, limited class labels, rendering most works unfeasible to employ unrestricted text descriptions for sound separation (Liu et al., 2023).

To address these challenges, we introduce an omni-modal sound separation model, OmniSep, to concurrently leverage omni-modal information. Specifically, we propose a `Query-Mixup` to mix up query features from different modalities, enabling OmniSep to optimize each modality concurrently and achieve a unified sound separation model capable of handling composed queries from diverse modalities. Expanding on this feature, we introduce the `negative query`, which identifies undesired sound information to eliminate specific sounds and enhance the flexibility of sound separation. Moreover, the reliance on predefined class labels in datasets confines current text-queried sound separation to predetermined categories, restricting the application of unrestricted text beyond the designated domain. To overcome this limitation, we propose `Query-Aug`, a retrieval-augmented method inspired by Lewis et al. (2020). This method retrieves the most similar in-domain class queries from the query set based on similarity as a reference query, facilitating open-vocabulary sound separation.

Our experimental results on MUSIC (Zhao et al., 2018), VGGSOUND-clean+ (Dong et al., 2022), and MUSIC-clean+ (Dong et al., 2022) datasets demonstrate the superior sound separation performance of our OmniSep across Text Query Sound Separation (TQSS), Image Query Sound Separation (IQSS), and Audio Query Sound Separation (AQSS) tasks, solidifying its position as an omni-modal sound separation model. OmniSep+`Neg Query` introduces information corresponding to unnecessary sounds, achieves the elimination of specific sounds, and enhances the flexibility and performance. Additionally, by adopting the Query-Aug strategy, the model's robustness to out-of-domain unrestricted text is improved, achieving good separation of open vocabularies. The code and models will be released, and the main contributions of this paper are as follows:

- We propose OmniSep, an omni-modal sound separation model that can separate sound based on queries of arbitrary modality, including single-modal queries such as text, images, and audio, as well as multi-modal composed queries.
- We introduce the negative query, leveraging undesired sound information information to filter out specific sound signals and further enhancing the flexibility of the sound separation model.
- We introduce `Query-Aug`, a retrieval augmented method, to achieve open-vocabulary sound separation, allowing querying with unrestricted natural language descriptions.
- Our experiments demonstrate that the OmniSep model achieves state-of-the-art sound separation performance across TQSS, IQSS, and AQSS.

## 2 RELATED WORKS

### 2.1 UNIVERSAL SOUND SEPARATION

The universal sound separation (Kavalerov et al., 2019; Liu et al., 2022; Pons et al., 2024) aims to extract distinct audio tracks from mixed audio, a critical task for audio understanding. Initially, research efforts were concentrated on specific domains such as speech (Wang & Chen, 2018; Luo et al., 2020; Li et al., 2023; Pegg et al., 2023; Wang et al., 2023b; Li et al., 2022) or music (Défossez et al., 2019; Manilow et al., 2022; Rouard et al., 2023; Luo & Yu, 2023) separation. Later, Kavalerov et al. (2019) employed permutation invariant training (PIT) (Yu et al., 2017) to separate mixed audio into multiple sound tracks of unidentified categories, yet remained limited to music, speech, and certain artificial sounds, lacking applicability to complex real-world sound understanding. To enhance comprehension of real-world audio, some researchers introduced extensive labeled audio datasets (Gemmeke et al., 2017; Chen et al., 2020), gradually achieving universal sound separation. Some (Ochiai et al., 2020; Kong et al., 2020; Liu et al., 2022) suggest utilizing class labels as queries for corresponding sound separation. MixIT (Wisdom et al., 2020) leverages a pre-trained sound classification model to conduct unsupervised training on unlabeled audio data. CLIPSEP (Dong et al., 2022) integrates visual data to enhance text query sound separation model training.

However, universal sound separation still faces two limitations: (1) Existing sound separation models are primarily trained on data with predefined class labels (Gemmeke et al., 2017; Chen et al., 2020), which restricts their ability to separate sounds based on out-of-domain text queries. (2) Previous separation methods have failed to utilize information related to interfering sounds, resulting in limited model performance and flexibility. To overcome these limitations, we introduce the `Query-Aug` strategy to enhance the robustness of sound separation models and enable open vocabulary sound separation. In addition, we propose the negative queries to handle undesired sound information, thereby enhancing the flexibility and performance of sound separation models.

## 2.2 QUERY-BASED SOUND SEPARATION

Query-based sound separation (QSS) (Ochiai et al., 2020; Kong et al., 2020; Liu et al., 2022; Dong et al., 2022) extracts specific audio tracks from mixed audio that match a given query. Previous research can be categorized into three main types: text query sound separation, image query sound separation, and audio query sound separation. Sound separation based on text queries (Ochiai et al., 2020; Kong et al., 2020; Liu et al., 2022) extracts relevant audio content from mixed sounds using textual descriptions. These methods are suitable for scenarios where a single text label can accurately describe the target sound. However, in more complex scenes such as outdoor environments where multiple sound sources are mixed, a single label may not suffice to describe the sound accurately. To address this, researchers (Tzinis et al., 2020; Dong et al., 2022; Chen et al., 2023) have utilized image as query to extract corresponding sounds. Moreover, certain sounds, like sound effects or abstract noises, are challenging to describe and may not be linked to visual content. Hence, researchers (Lee et al., 2019; Chen et al., 2022) have proposed audio queried methods to separate such abstract sounds.

However, existing sound separation models are typically tailored for single-modal queries and encounter difficulties in separating sound with composed queries. To tackle this limitation, we introduce the query-mixup training strategy, which mixes up the features from different modalities during training. This allows OmniSep to unify the training objectives and optimize each modality concurrently, thereby facilitating sound separation based on omni-modal query.

## 2.3 MULTI-MODALITY REPRESENTATION LEARNING

In recent years, numerous studies (Girdhar et al., 2023; Wang et al., 2023a; 2024d;b;c; Zhang et al., 2025) have employed self-supervised learning to establish alignment across multiple modalities. Initially, CLIP (Radford et al., 2021) was trained on extensive public data to construct alignment between images and texts. Subsequently, CLAP (Elizalde et al., 2023) adopted a similar methodology to learn alignment between audio and text modalities. However, these works only focused on alignment between two modalities. On the other hand, Imagebind (Girdhar et al., 2023) aligns diverse modalities individually to a unified image modality, thereby enabling the unified alignment of multiple modalities, including text, images, audio, depth, and more. Consequently, numerous studies (Cheng et al., 2023a;b;c; Huang et al., 2023; Lei et al., 2024; Fu et al., 2024) have emerged to investigate various multimodal tasks leveraging these unified representations. For instance, Xu et al. (2023); Liang et al. (2023) achieve the open vocabulary segmentation with the joint text-image representations, while Han et al. (2023) expands upon this groundwork to enhance multimodal comprehension. Meanwhile, Dong et al. (2022) delves into the domain of zero-shot text-guided sound separation with the aid of image guidance.

Despite the rapid advancements in multi-modal representation learning, there is currently no dedicated multi-modal unified model tailored for sound separation tasks capable of handling text, audio, and image modal queries. In response, we introduce the first omni-modal sound separation model.

## 3 METHODS

### 3.1 OVERVIEW

The Omni-modal sound separation model (**OmniSep**) aims to perform sound separation based on different modal queries, where the query can originate from the audio modality ($A$), text modality ($T$), or visual modality ($V$). It separates noisy mixed audio into clean audio consistent with the query, as described in Section 3.2. Additionally, in Section 3.3, we introduce a method to incorporate

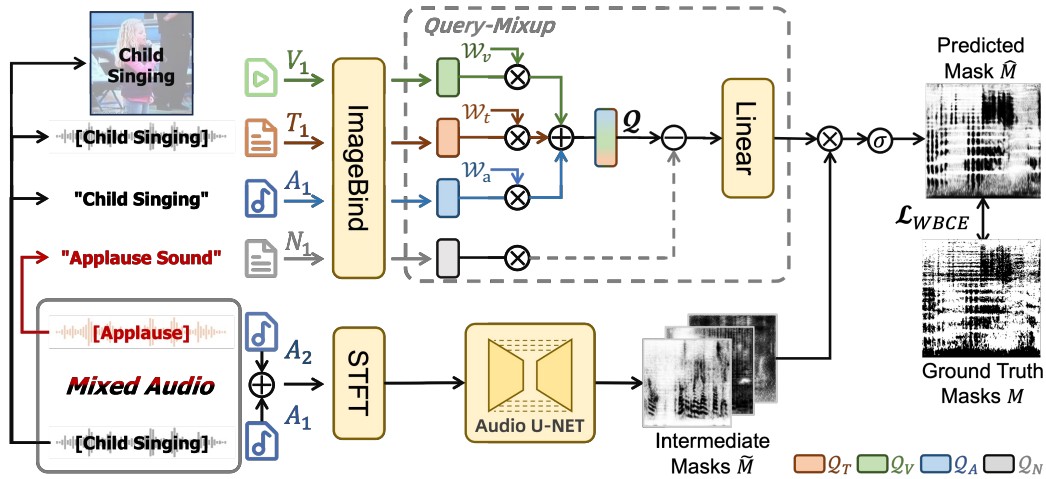

Figure 1: Illustration of Omni-modal Sound Separation (**OmniSep**). The OmniSep employs the parameter-frozen ImageBind model to extract features from diverse modal queries, denoted as $\mathbf{Q}_T$, $\mathbf{Q}_V$, and $\mathbf{Q}_A$ within the figure. Negative query $N_1$, which aligns semantically with the interference audio $A_2$, is adopted to aid sound separation during inference. Note that during testing time for IQSS, TQSS, and AQSS, only a single modal query is employed.

interference noise information into the model based on the composed query of the original query and the negative query to assist the model in sound separation. To further achieve open vocabulary sound separation, we propose `Query-Aug` in Section 3.4, which allows the model to querying with any unrestricted natural language description.

## 3.2    OMNISEP: OMNI-MODAL SOUND SEPARATION.

OmniSep adopts an architecture similar to CLIPSEP (Dong et al., 2022), as illustrated in Figure 1. The Query-Net converts data from various modalities into corresponding query features $\mathbf{Q} \in \mathbb{R}^{n \times 1024}$, where $n$ denotes the number of audio samples in the mixed audio. Subsequently, we employ the Separate-Net to transform the audio mixture $A_{\mathrm{mix}}$, consisting of $n$ audio sources, into clean sound $A_{\mathrm{clean}}$ based on the query features $\mathbf{Q}$. However, since CLIPSEP periodically switches the query modality during training, the training objective remains unfixed, posing challenges in achieving optimal performance across all modalities. We address this issue using the `Query-Mixup` strategy.

**Query-Net**   To train a unified sound separation model adaptable to queries of various modalities, we selected the imagebind model[1], pre-trained on multiple modalities, to extract query features: $\mathbf{Q}_A, \mathbf{Q}_V, \mathbf{Q}_T = \texttt{ImageBind}(A, V, T)$ where $\mathbf{Q}_A, \mathbf{Q}_V, \mathbf{Q}_T \in \mathbb{R}^{n \times 1024}$, with the model parameters kept frozen during training. Subsequently, we adopt the `Query-Mixup` strategy to mix up query features from different modalities, enabling OmniSep to optimize each modality concurrently and unify the training objectives with randomly sampled weight factors $w_a$, $w_v$, and $w_t$:

$$\mathbf{Q} = \frac{w_a \mathbf{Q}_A + w_v \mathbf{Q}_V + w_t \mathbf{Q}_T}{w_a + w_v + w_t}, \quad w_a, w_v, w_t \in [0, 1]. \tag{1}$$

**Separate-Net**   For a mixed audio signal $A_{\mathrm{mix}}$ composed of $n$ audio sources $\{A_1, A_2, \cdots, A_n\}$, each audio source $A_i$ is associated with corresponding video $V_i$ and textual query $T_i$, forming $n$ triplets $\{(A_1, V_1, T_1), \cdots, (A_n, V_n, T_n)\}$. The query $\mathbf{Q}_i$ for sound separation is derived from the $i$-th triplet $((A_i, V_i, T_i)$. The mixed audio is first converted into the magnitude spectrum $X \in \mathbb{R}^{C \times F \times T}$ using the Short-Time Fourier Transform (STFT), where $C$ represents the number of channels, $F$ is the frequency dimension and $T$ is the time dimension. For single-channel audio in this paper, $C = 1$. Subsequently, we input the spectrogram into the audio U-Net (Ronneberger et al., 2015; Jansson et al., 2017) and obtain $k$ intermediate masks $\tilde{M} = \{\tilde{M}_1, \cdots, \tilde{M}_j, \cdots, \tilde{M}_k\}$, where $\tilde{M} \in \mathbb{R}^{k \times F \times T}$, $\tilde{M}_j$ is the $j$-th intermediate mask and $k \geq n$. In this work, following the setting of CLIPSEP (Dong et al., 2022), $k$ is set to 32. At the same time, each query feature $\mathbf{Q}_i$ obtained from the $(A_i, V_i, T_i)$ is

---

[1]https://github.com/facebookresearch/ImageBind

transformed into $k$-dimensional channel-wise weight $q_i$ through the fully connected mapping layer: $q_i = \text{Linear}(\mathbf{Q}_i)$, where $q_i \in \mathbb{R}^k$. Finally, the final predicted masks $\hat{M}_i \in \mathbb{R}^{C \times F \times T}$ corresponding to $A_i$ can be obtained with the channel-wise weight $q_i$ and the intermediate masks $\tilde{M}$:

$$\hat{M}_i = \sum_{j=1}^{k} \sigma(w_{ij} q_{ij} \tilde{M}_j + b_i), \tag{2}$$

where $w_i \in \mathbb{R}^k$ is a learnable scale vector, $b_i \in \mathbb{R}$ a learnable bias, $q_{ij}$ is the channel-wise weight of the $j$-th intermediate mask $\tilde{M}_j$, and $\sigma(\cdot)$ the sigmoid function. The training objective of the entire model is the sum of the weighted binary cross entropy (WBCE) losses for each query source:

$$\mathcal{L} = \sum_{i=1}^{n} \text{WBCE}(M_i, \hat{M}_i) = \sum_{i=1}^{n} X \odot \left( -M_i \log \hat{M}_i - (1 - M_i) \log(1 - \hat{M}_i) \right), \tag{3}$$

where $M_i$ is the ground truth mask for audio source $A_i$. The predicted magnitude spectrum is first obtained by applying the predicted mask $\hat{M}_i$ to the magnitude spectrum of the mixed audio. Finally, the noisy phase is combined with the predicted magnitude, and the inverse short-time Fourier transform (iSTFT) is applied to convert the spectrogram back to a time-domain waveform, resulting in the separated audio signal. During inference, it's important to note that for the sound separation of single-modal queries, we input the single-modal query into the Sep-net without employing the Query-Mixup strategy. Furthermore, the subsequent subsections introduce two training-free strategies for query operations aimed at further enhancing the performance of the sound separation model.

### 3.3 NEGATIVE QUERY: ELIMINATE INTERFERENCE INFORMATION FROM THE ORIGINAL QUERY.

For noise information, we treat it as a novel form of query and extract it into the corresponding Query feature $\mathbf{Q}_N$, employing the same method as for other modal queries. With the aid of multi-modal pre-training (Girdhar et al., 2023) and the `Query-Mixup`, features linked to different modal queries can be adeptly mapped into the same space, facilitating sound separation. Drawing inspiration from multi-modal joint retrieval (Wang et al., 2024a), we employ the vector subtraction technique to eliminate the information associated with $\mathbf{Q}_N$ from the original Query $\mathbf{Q}$. However, it's worth noting that compared to retrieval tasks, sound separation tasks demand a more fine-grained manipulation of representations. Direct subtraction may lead to the loss of local information in $\mathbf{Q}$, where any local information may be linked to sounds on certain frequency bands. Thus, when employing negative queries, we aim to maintain the stability of the frequency bands corresponding to non-key content by introducing a negative query weight $\alpha$, which is expressed as:

$$\mathbf{Q}' = (1 + \alpha)\mathbf{Q} - \alpha \mathbf{Q}_N. \tag{4}$$

Subsequently, the final query feature $\mathbf{Q}'$ is inputted into the Separate-Net for sound separation. In particular, when $\alpha = 0$, $\mathbf{Q}' = \mathbf{Q}$, which means that no noise information is applicable during sound separation.

### 3.4 QUERY-AUG: TOWARDS THE QUERY OF UNRESTRICTED NATURE LANGUAGE DESCRIPTIONS.

Given that the majority texts in current audio-text paired datasets are accompanied by predefined category labels, the sound separation model encounters challenges when confronted with more authentic and unrestricted natural language descriptions during inference. To tackle this issue, we introduce the `Query-Aug` method, which facilitates open-vocabulary sound separation. In this method, a query set (Query-Set $\in \mathbb{R}^{M \times 1024}$) is constructed, comprising query features extracted using ImageBind for each class label, where $M$ represents the number of class labels utilized during training. For an unrestricted natural language description, the query feature $\mathbf{Q}_{aug} \in \mathbb{R}^{1024}$ in the set that is most closely related to its corresponding ImageBind query feature $\mathbf{Q}_{des} \in \mathbb{R}^{1024}$ is identified:

$$\mathbf{Q}_{aug} = \underset{\mathbf{Q}m \in \text{Query-Set}}{\arg\max} \cos(\mathbf{Q}_{des}, \mathbf{Q}_m), \tag{5}$$

where $\cos$ represents the cosine similarity between the query features. Subsequently, $\mathbf{Q}_{aug}$ serves as the query for Separate-Net, which uses it to generate the corresponding separated audio signals.

# 4 EXPERIMENTS

## 4.1 IMPLEMENTATION DETAILS

We conducted experiments on two datasets, VGGSOUND (Chen et al., 2020) and MUSIC (Zhao et al., 2018), to evaluate the sound separation effect under different modal queries. VGGSOUND stands out as the largest audio-video consistent sound description dataset, boasting 550 hours of audio across 330 different sound categories, thereby representing various sound separation scenarios. Conversely, MUSIC focuses on instrument sounds, providing 10 hours of recordings featuring diverse instruments alongside corresponding performance videos, thus delineating a specific domain within sound separation. For comparison with previous studies, we conducted sound separation experiments following the CLIPSEP. This entailed training our model on the MUSIC dataset and evaluating its performance on itself. Additionally, we trained the model on the VGGSOUND dataset and evaluated its performance on the VGGSOUND-CLEAN+ and MUSIC-CLEAN+ datasets, which contain manually processed clean sound separation evaluation samples. For more details about the dataset, please refer to Appendix D.

Following the experimental settings of CLIPSEP, a 4-second audio segment is randomly selected from the entire audio as the audio source. For queries, we extract the audio feature of the entire audio as the audio query; the average image feature of 4 frames with a 1-second interval from the corresponding video serves as the image query, and the class label associated with the audio as the text query. Note that, during the inference, we extract $\mathcal{S}$ audio samples from the training set for each class as its audio query (where $\mathcal{S} = 5$ for VGGSOUND, and $\mathcal{S} = 1$ for MUSIC) according to the setting of (Lee et al., 2019), for audio query sound separation (AQSS). More implementation details are shown in the Appendix A.

## 4.2 MAIN RESULTS

**OmniSep: Sound Separation for Omni-modal Queries.** The OmniSep is designed to address sound separation tasks with diverse modal queries. Table 1 presents the SDR comparison of sound separation across various modal queries. Specifically, TQSS, IQSS, and AQSS represent text-, image-, and audio-queried sound separation, respectively, while *Non-Queried Models* denotes the method of implicitly learning sounds within the model. Across each task in TQSS, IQSS, and AQSS, our OmniSep achieves state-of-the-art sound separation performance, improving the Mean SDR by 0.43 to 4.36. This demonstrates that OmniSep can be queried with any single modality query. Furthermore, when queried with a composed Omni-modal query, OmniSep can refer to information from different modalities to achieve a more robust sound separation performance, with the Mean SDR of 7.46 on VGGSOUND-CLEAN+. Additionally, negative queries further enhance the sound separation performance by leveraging noise information during test-time. Compared to OmniSep without negative queries, the Mean SDR is further enhanced by 0.10 to 1.60. A detailed analysis of negative queries is provided in subsection 4.3.

**Query-Mixup: Enhancing Multi-modal Joint Training.** To further validate the effectiveness of our proposed method, we conducted detailed ablation experiments as shown in Table 2: **(1) Multi-modal joint training for different modal queried sound separation**: The comparison between methods trained using single-modal data and multi-modal data highlights the adaptability of the latter in handling sound separation tasks with different modal queries. While the single-modality training method may yield better results on specific tasks (e.g., #1 trained with only text queries achieves a 0.73 higher Mean SDR on TQSS than #3), the #3 model, jointly trained with multi-modal queries, achieves an AVG SDR of 6.42, consistently outperforming #1 and #2 with improvements of $0.39 \sim 1.30$. Furthermore, the introduction of audio modality data during training (#4) leads to further performance enhancement, with a 0.03 increase in performance from #3 to #4. **(2) Query-mixup for unified omni-modal sound separation**: To handle sound separation tasks with various modal queries, we employ `Query-Mixup`. In Experiment #5, we observed that the average SDR of 6.70 surpassed Experiment #4 of 6.45 by 0.25. Remarkably, this enables us to achieve comparable sound separation performance in TQSS as the text-specific training method (#1). This indicates that the `Query-Mixup` method circumvents the problem of fluctuating training objectives in traditional multimodal joint training methods, which often affects the performance of single-modal queried sound separation.

Table 1: Comparison of sound separation performance among different methods on **MUSIC**, **MUSIC-CLEAN+**, and **VGGSOUND-CLEAN+**. Experiments marked with * are reproduced using CLIPSEP's dataset partitioning for comparison. All **+NQ** results are outcomes when $\alpha = 0.5$.

| Method | MUSIC | | VGGSOUND-CLEAN+ | | MUSIC-CLEAN+ | |
|---|---|---|---|---|---|---|
| | Mean SDR | Med SDR | Mean SDR | Med SDR | Mean SDR | Med SDR |
| *Non-Queried Sound Separation* | | | | | | |
| LabelSep | 8.18±0.80 | 7.82 | 5.55±0.81 | 5.29 | - | - |
| PIT (Kavalerov et al., 2019) | 8.68±0.76 | 7.67 | 5.73±0.79 | 4.97 | 12.24±1.20 | 12.53 |
| TDANet (Li et al., 2022) | **10.31±0.79** | 10.18 | **6.43±0.62** | **5.83** | **13.09±1.11** | 12.74 |
| *Text-Queried Sound Separation* | | | | | | |
| BERTSep | - | - | 5.09±0.80 | 5.49 | 4.67±0.44 | 4.41 |
| CLIPSEP-NIT (Dong et al., 2022) | - | - | 3.05±0.73 | 3.26 | 10.27±1.04 | 10.02 |
| CLIPSEP (Dong et al., 2022) | 8.36±0.83 | 8.72 | 2.76±1.00 | 3.95 | 9.71±1.21 | 8.73 |
| CLIPSEP-Text (Dong et al., 2022) | 7.91±0.81 | 7.46 | 5.49±0.82 | 5.06 | 10.73±0.99 | 9.93 |
| AudioSEP (Liu et al., 2023)* | 9.82±0.89 | 8.76 | 6.26±0.87 | 5.57 | 11.23±0.99 | 10.28 |
| OmniSep(ours) | 10.65±1.07 | 9.97 | 6.70±0.66 | 5.73 | 12.55±0.77 | 12.68 |
| OmniSep(ours)+NQ | **10.92±1.06** | 9.97 | **7.57±0.67** | **6.52** | **14.15±0.95** | **14.46** |
| *Image-Queried Sound Separation* | | | | | | |
| SOP (Zhao et al., 2018) | 6.59±0.85 | 6.22 | 2.99±0.84 | 3.89 | 11.44±1.18 | 11.18 |
| CLIPSEP (Dong et al., 2022) | 8.06±0.79 | 8.01 | 5.46±0.79 | 5.35 | 12.20±1.17 | 12.42 |
| i-Query (Chen et al., 2023)* | 10.54±0.72 | 9.73 | - | - | - | - |
| OmniSep(ours) | 10.97±1.03 | 10.21 | 6.69±0.67 | 6.43 | 13.15±0.92 | 13.89 |
| OmniSep(ours)+NQ | **11.08±1.02** | **10.22** | **7.68±0.69** | **6.60** | **13.78±1.02** | **13.99** |
| *Audio-Queried Sound Separation* | | | | | | |
| AQSS (Lee et al., 2019)* | 6.43±0.92 | 5.73 | 5.34±0.71 | 5.30 | 8.56±0.75 | 7.82 |
| OmniSep(ours) | 10.26±1.13 | 10.06 | 7.12±0.65 | 5.45 | 12.92±0.89 | 14.04 |
| OmniSep(ours)+NQ | **10.40±1.11** | **10.14** | **7.22±0.68** | 5.07 | **13.43±0.98** | **14.14** |
| *Composed Omni-Modal Queried Sound Separation* | | | | | | |
| OmniSep(ours) | 11.03±1.05 | 10.21 | 7.46±0.65 | 6.32 | 13.49±0.94 | 14.04 |
| OmniSep(ours)+NQ | **11.16±1.04** | **10.33** | **8.00±0.69** | **6.43** | **13.82±0.98** | **14.07** |

Table 2: Comparison of SDR values on TQSS and IQSS among sound separation models trained with diverse modality data (**Text**, **Image** and **Audio**) and training methods (**MixUP**). **AVG SDR** represents the average SDR across different sound separation models queried with text and image.

| ID | Text | Image | Audio | MixUP | TQSS | | IQSS | | AVG SDR |
|---|---|---|---|---|---|---|---|---|---|
| | | | | | Mean SDR | Med SDR | Mean SDR | Med SDR | |
| #1 | ✓ | | | | **6.70±0.68** | 5.81 | 3.53±0.49 | 2.86 | 5.12±0.59 |
| #2 | | ✓ | | | 5.72±0.83 | 5.41 | 6.33±0.68 | 5.74 | 6.03±0.76 |
| #3 | ✓ | ✓ | | | 6.33±0.71 | 5.83 | 6.51±0.72 | 5.77 | 6.42±0.72 |
| #4 | ✓ | ✓ | ✓ | | 6.37±0.68 | 5.68 | 6.53±0.68 | 6.33 | 6.45±0.68 |
| #5 | ✓ | ✓ | ✓ | ✓ | **6.70±0.66** | **6.22** | **6.69±0.67** | **6.43** | **6.70±0.66** |

## 4.3 NEGATIVE QUERY: ENHANCING SOUND SEPARATION WITH NOISE INFORMATION.

To further enhance the efficacy of sound separation, we introduce negative queries to leverage noise information. Illustrated in Figure 2 are the variations of SDR with the negative query weight $\alpha$ on VGGSOUND-CLEAN+ and MUSIC-CLEAN+. We compare two distinct methods of negative query augmentation: firstly, the naive subtraction method ($\mathbf{Q}' = \mathbf{Q} - \alpha\mathbf{Q}_N$), which employs negative representation processing akin to retrieval tasks; and secondly, our proposed method ($\mathbf{Q}' = (1+\alpha)\mathbf{Q} - \alpha\mathbf{Q}_N$), which additionally enhances the original query with proportionally weight. Please note that in both methods, when $\alpha = 0$, the negative query is not adopted.

Here are the conclusions drawn from the figure: **(1) Adaptability of Negative Query**: Across all sound separation tasks (TQSS, IQSS, and AQSS), the negative query significantly enhances sound

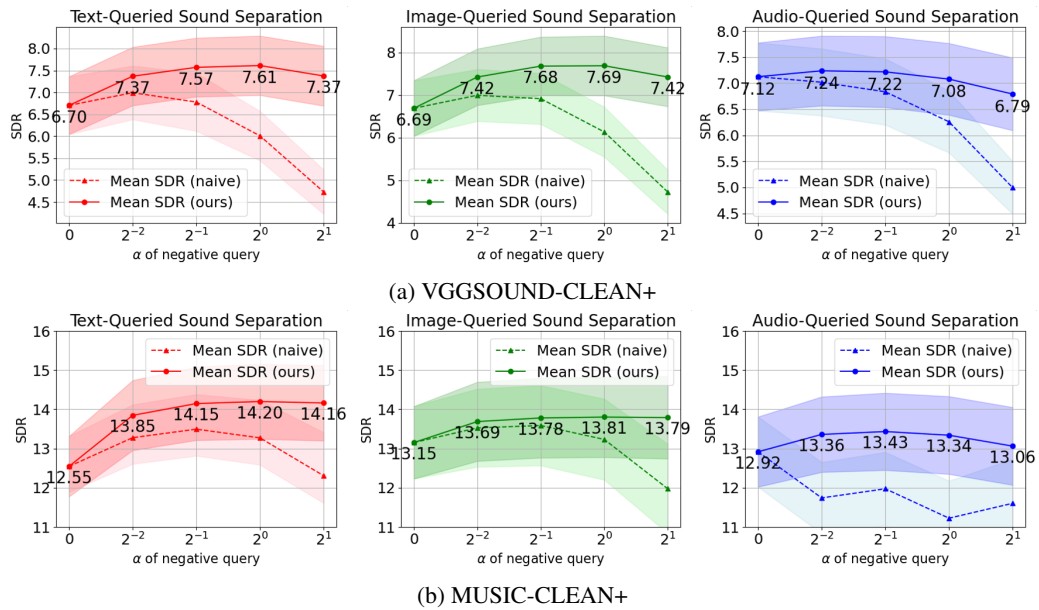

(a) VGGSOUND-CLEAN+

(b) MUSIC-CLEAN+

Figure 2: The variation of SDR with the negative query weight $\alpha$ on VGGSOUND-CLEAN+ and MUSIC-CLEAN+. The x-axis represents the weight $\alpha$ of negative query, while the y-axis denotes the SDR. The shaded area indicates the standard deviation of the SDR. The dashed and solid lines respectively represent the results of naive subtraction ($\mathbf{Q}'=\mathbf{Q}-\alpha\mathbf{Q}_N$) and ours ($\mathbf{Q}'=(1+\alpha)\mathbf{Q}-\alpha\mathbf{Q}_N$).

separation performance. For instance, on MUSIC-CLEAN+, when $\alpha = 1$, the Mean SDR stands at 14.20, marking a notable improvement of 1.65 from the 12.55 achieved without negative queries. **(2) Proportional Weighting vs. Direct Subtraction**: Comparative analysis reveals that our proposed proportional weighting method consistently outperforms traditional direct subtraction method. Across all tasks and varying alpha values, the solid line representing our method consistently surpasses the performance depicted by the dotted line, which represents direct subtraction. This demonstrates the efficacy of our proportional weighting approach in mitigating interference on the original query information when negative queries are employed. **(3) Robustness of Weight Selection**: As a training-free method, it offers considerable flexibility in utilizing negative queries during the inference process, allowing for the use of any weight $\alpha$ to negative query. However, this flexibility also presents challenges in determining the optimal weight $\alpha$. The Mean SDR of naive direct subtraction shows significant fluctuations with changes in the weight $\alpha$, highlighting the considerable impact of $\alpha$ on the naive method. Moreover, the optimal $\alpha$ value for achieving the best performance for naive direct subtraction varies across different tasks and datasets. Hence, finding a universally applicable fixed weight $\alpha$ for the naive direct subtraction method is not feasible. On the contrary, even across various tasks and datasets, the mean SDR range resulting from the introduction of negative queries using our proposed proportional weighting method is no more than 0.45 for different $\alpha$ weights, thus obviating the need for excessive adjustment of the weight $\alpha$. (For the TQSS task on VGGSOUND-CLEAN+, when $\alpha$=2, the Mean SDR of our method remains at 7.37, while the naive sound separation method has already decreased to 4.32), demonstrating significant robustness.

## 4.4 Sound Separation with Unrestricted Textual Descriptions.

As mentioned in Liu et al. (2023), previous methods (Ochiai et al., 2020; Veluri et al., 2023) for text-queried sound separation relied on predefined class labels as queries, constraining their adaptability to accommodate unrestricted natural language descriptions. To evaluate sound separation performance with such unrestricted descriptions as queries, we utilized GPT-3.5 to rewrite all predefined class labels into varied yet semantically consistent textual descriptions. Further details are provided in Appendix B.

In Table 3, we conduct a comparative analysis of the sound separation performance achieved by various methods using different text queries: (1) **Predefined tags vs. Unrestricted descriptions**: Training on predefined class labels limits the model's ability to generalize to out-of-domain text. A

significant drop in performance is observed when inferring out-of-domain text (Mean SDR drops from 5.49 in Experiment #6 to 3.53 in Experiment #8). **(2) Multi-modal joint training vs. Single-modal training**: Joint training with multi-modal queries (Experiment #9) helps fill the gap between the limited representations of text queries, enhancing generalization to out-of-domain text. **(3) Open-vocabulary sound separation with Query-Aug**: Despite the improvement in robustness achieved by multi-modal joint training, a performance gap remains due to cross-modal differences. The Mean SDR of 4.95 in Experiment #10 is significantly lower than that of Experiment #6,

which achieved a sound separation performance of 5.49 with in-domain class labels. Our proposed `Query-Aug` method addresses this by selecting suitable in-domain text for text enhancement based on text similarity. The `Query-Aug` method significantly enhances the robustness of the sound separation model against out-of-domain text. When employing unrestricted out-of-domain text descriptions for sound separation, the Mean SDR is further improved by 1.37 compared to OmniSep (Experiment #11 yields an SDR of 6.32, whereas Experiment #10 achieves 4.95). It's noteworthy that the performance of OmniSep+`Query-Aug` (#11) with out-of-domain text is comparable to that achieved

Table 3: The performance comparison of sound separation between queries using predefined class labels and unrestricted textual descriptions on the VGGSOUND-CLEAN+ dataset.

| ID | Method | Mean SDR | Med SDR |
|---|---|---|---|
| *Query with predefined class labels.* | | | |
| # 6 | CLIPSEP-Text | 5.49±0.82 | 5.06 |
| # 7 | OmniSep | **6.70±0.66** | **5.73** |
| *Query with unrestricted textual descriptions.* | | | |
| # 8 | CLIPSEP-Text | 3.53±0.52 | 2.91 |
| # 9 | +Query-Aug | 5.24±0.79 | 4.87 |
| # 10 | OmniSep | 4.95±0.81 | 4.45 |
| # 11 | +Query-Aug | **6.32±0.64** | **5.97** |

with predefined in-domain class labels (#6, #7). Specifically, while Experiment #6 achieves a Mean SDR of 5.49, Experiment #11 attains a Mean SDR of 6.32, which is a superior performance despite being in a more challenging condition, achieving open-vocabulary sound separation.

## 4.5 How Query-Mixup Enables Omni-Modal Sound Separation?

Figure 3 illustrates the spatial distribution relationship of imagebind representations across different modalities, showing that features from various modalities cluster independently with significant cross-modal gaps, as discussed in Liang et al. (2022). This makes it challenging for a single sound separation model to process data from different modalities simultaneously. In previous methods, such as CLIPSEP, the model constantly switches between text and image features, making it difficult to simultaneously accommodate the distinct distribution requirements of both text and image representations. Consequently, the model fails to achieve optimal performance for both modalities at the same time.

Our proposed OmniSep uses a query-mixup strategy to combine features of the same semantics from different modalities, effectively bridging the gap between their distributions with "mixed embeddings". It can optimize the mixed embeddings, composed of different modal embeddings, to accommodate both uni-modality queries and multi-modality composed queries. In omni-modal composed sound separation, queries from different modalities

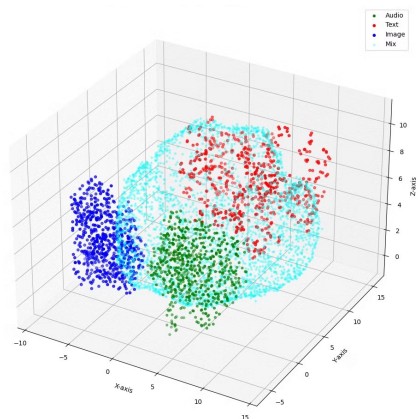

Figure 3: UMAP visualization of three different modal imagebind embeddings. The mix embedding is a weighted embedding of the three modalities.

correspond to the same semantics and can complement each other, ensuring the model has a more comprehensive understanding of the query information, thereby improving sound separation.

In particular, functions such as omni-modal composed sound separation and negative query rely on the additive and subtractive characteristics of embeddings between different modalities. The query embeddings required for composed-query and negative-query are all in the mixed embedding area between modalities. Therefore, the `Query-Mixup` strategy forms the basis of composed-query and negative query capabilities, enhancing the model's controllability.

Table 4: Qualitative results of sound separation using different modal queries. **Interference** and **Target** denote the reconstructed signals of interference audio and ground truth audio using the ground truth ideal binary masks, respectively. All spectra are presented on a logarithmic frequency scale.

| Query | Mixture | Interference | Target | Prediction |
|---|---|---|---|---|
| *Child Singing* | | | | |
| | | | | |
| | | | | |

## 4.6 QUALITATIVE ANALYSIS

To demonstrate the effectiveness of OmniSep, we conduct a qualitative analysis of sound separation with various modal queries, as depicted in Table 4. The results showcase our OmniSep successfully separate audio signals corresponding to the queries across TQSS, IQSS, and AQSS tasks. Additionally, in Appendix C.1, we present qualitative results of sound separation with and without negative queries in TQSS. It's noteworthy that in a sample of child singing audio mixed with child crying noise interference, traces of the crying persist in the separation results, posing challenges to effective separation due to the association of both sounds with the child. However, results obtained with negative queries demonstrate that OmniSep+Negative Query can effectively eliminate noise by leveraging the undesired sound information. The introduction of negative query information significantly enhances the removal of interfering sound signals and improves the flexibility of the sound separation model. In addition, as demonstrated in the comparison of sound separation using unrestricted natural language queries in Appendix C.2, `Query-Aug` enhances the model's ability to generalize to out-of-domain unrestricted descriptions, thereby achieving open-vocabulary sound separation. Please visit the demo page at `https://omnisep.github.io/` to see more sound separation results and learn more about OmniSep.

## 5 CONCLUSION

Researchers have adopted sound separation to scaling up audio datasets, but existing sound separation methods are limited to single-modal queries and have certain limitations. In this study, we introduce the multi-modal query mixing strategy, `Query-Mixup`, and propose the first Omni-Modal Sound Separation model (OmniSep). OmniSep can perform sound separation based on any modal query, including single-modal queries such as text, images, and audio, as well as multi-modal composed queries. Additionally, we propose a method to remove undesired sound information according to negative queries from the original query, thereby improving sound separation performance and model flexibility. To overcome the challenge of limited text labels, we introduce `Query-Aug`, achieving open-vocabulary sound separation and facilitating the use of unrestricted natural language queries. Experimental results on different modal sound separation tasks (TQSS, IQSS, AQSS) demonstrate that our model achieves state-of-the-art performance in omni-modal sound separation.

ACKNOWLEDGMENTS

This work was supported in part by National Natural Science Foundation of China under Grant No. 62222211 and No.624B2128. Thank Ziyang Ma and Slytherin Wang for their insightful discussions and valuable suggestions regarding this project.

REPRODUCIBILITY STATEMENT

All our code, and model weights will be open-sourced. In Section 3, we provide a detailed description of the OmniSep modules and training instructions. Section 4.1 and Appendix A offer additional details on the training process.

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

## A    IMPLEMENTATION DETAILS

Same as the experimental setting of Dong et al. (2022), for all audio samples, we conducted experiments on samples of length 65535 (approximately 4 seconds) at a sampling rate of 16 kHz. For spectrum computation, we employed a short-time Fourier transform (STFT) with a filter length of 1024, a hop length of 256, and a window size of 1024. All images were resized to $224 \times 224$ pixels. The audio model in this paper is a wildly used 7-layer U-Net network with $k = 32$, generating 32 intermediate masks. All models were trained with a batch size of 128, using the Adam optimizer with parameters $\beta_1 = 0.9$, $\beta_2 = 0.999$, and $\epsilon = 10^{-8}$, for 200,000 steps. Additionally, we employed warm-up and gradient clipping strategies, following Dong et al. (2022). We compute the signal-to-distortion ratio (SDR) using museval (Stöter et al., 2018). All experiments were conducted on a single A800 GPU.

## B    SAMPLE OF UNRESTRICTED NATURAL LANGUAGE DESCRIPTIONS.

We utilized ChatGPT 3.5 to transform each class label from VGGSOUND into natural language descriptions without constraints, ensuring semantic consistency. Table 5 displays the instruction prompt employed in ChatGPT. For further analysis of the Query-Aug, Table 6 presents numerous examples of these unrestricted descriptions.

Table 5: Instruction prompt when using chatgpt to generate unlimited natural language descriptions.

| | |
|---|---|
| **User:** | Could you please assist me in rephrasing these descriptions of audio content to convey the same meaning but with different wording? |
| **Chatgpt:** | Of course! Please provide the descriptions you'd like me to rephrase, and I'll help you with alternative wording. |
| **User:** | tapping guitar |
| **Chatgpt:** | Tapping on a guitar to produce rhythms |
| **User:** | dog barking |
| **Chatgpt:** | A dog vocalizing with a bark. |

Table 6: Samples of class labels along with corresponding unrestricted natural language descriptions.

| **Predefined Class Label** | **Unrestricted Nature Language Description** |
|---|---|
| playing didgeridoo | Performing on a didgeridoo instrument. |
| golf driving | Swinging a golf club on the course. |
| dog barking | A dog letting out barking noises. |
| playing timpani | Beating drums in a timpani performance. |
| magpie calling | The melodious call of a magpie bird. |
| dog bow-wow | A dog letting out a woofing sound. |
| subway metro underground | The rumbling noise of a subway train passing. |
| cat growling | A cat emitting a growling noise. |

## C    MORE EXPERIMENTS

### C.1    QUALITATIVE ANALYSIS ON SOUND SEPARATION WITH NEGATIVE QUERY.

In Table 7, we provide a comparison of the sound separation effect between employing negative query (OmniSep+Neg query) and not using it (OmniSep). We exclusively showcase TQSS as an illustrative example, acknowledging the consistent performance of negative query across different modal queries in sound separation tasks. As illustrated in the Mel spectrogram comparison of the separation results, integrating negative query significantly enhances the removal of interfering audio signals from the mixed audio, effectively isolating the content associated with the negative query. Additional examples and corresponding audios are available on the demo page. Comparing the raw files of the two methods, it's evident that using negative query can greatly enhance the flexibility of the sound separation model, particularly when addressing challenges such as the inability to effectively differentiate between two sound signals using a single text query.

Table 7: Qualitative comparison of sound separation models with and without negative queries. All **OmniSep+Neg Query** experiments are obtaned with $\alpha = 1$. We highlighted the enhanced separation effect achieved by employing negative queries with red boxes.

| Mixture | Interference | Target | OmniSep | OmniSep+Neg Query |
|---------|--------------|--------|---------|-------------------|

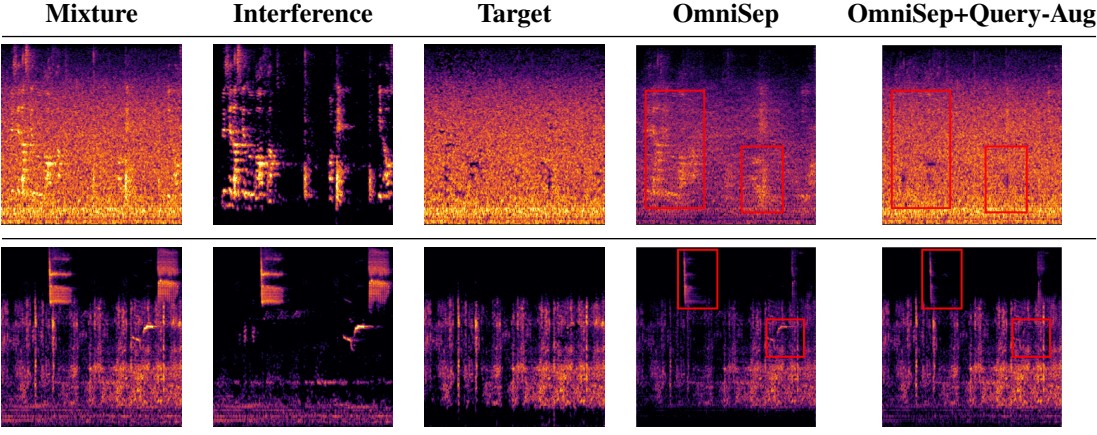

Table 8: Qualitative comparison of sound separation models with and without negative queries. We highlighted the enhanced separation effect achieved by employing negative queries with red boxes.

| Mixture | Interference | Target | OmniSep | OmniSep+Query-Aug |
|---------|--------------|--------|---------|-------------------|

## C.2 QUALITATIVE ANALYSIS ON SOUND SEPARATION WITH QUERY-AUG.

In Table 7, we display the sound separation outcomes for an unrestricted natural language description query. It's clear from the mel spectrogram that when the query-augmentation method is not employed, the model (OmniSep) faces difficulties in grasping the sound information linked to the query. This results in a reduced capability to effectively separate the sound signals associated with the unrestricted natural language description. Conversely, with the integration of the query-augmentation method, the model (OmniSep+Query-Aug) showcases enhanced effectiveness in segregating the corresponding sound information content and removing interfering sound signals.

## C.3 OPEN VOCABULARY SOUND SEPARATION ON DIVERSE EVALUATION SETS.

To further demonstrate the effectiveness of our proposed Query-Aug method for open-vocabulary sound separation, we integrated it with AudioSep and evaluated its performance on the datasets you suggested. The results are presented in Table 9. It is important to note that AudioSet and VGGSOUND use class labels as queries, which are in-domain queries already encountered during training. Consequently, when applying Query-Aug to these datasets, the retrieved query remains identical to the original, resulting in no performance change. Therefore, we did not include evaluations on AudioSet and VGGSOUND. Across other datasets, including AudioCaps, MUSIC, ESC-50, and

Table 9: Performance comparison of open-vocabulary sound separation on diverse evaluation sets. For a fair evaluation, we compare the results of AudioSep and AudioSep + Query-Aug.

| Method | AudioCaps | | Music | | ESC-50 | | Clotho | |
|---|---|---|---|---|---|---|---|---|
| | SDRi | SI-SDR | SDRi | SI-SDR | SDRi | SI-SDR | SDRi | SI-SDR |
| AudioSep | 7.68 | 6.45 | 9.75 | 8.45 | 10.24 | 9.16 | 6.51 | 4.84 |
| +Query-Aug(ours) | **8.22** | **7.19** | **10.72** | **9.71** | **10.72** | **9.84** | **7.20** | **5.80** |

Clotho, Query-Aug led to significant performance improvements, demonstrating its effectiveness in enhancing open-vocabulary capabilities.

### C.4 THE ABLATION PERFORMANCE ON AQSS

To provide a more comprehensive comparison, we present the omitted AQSS results in Table 10, which were excluded from Table 2 due to space constraints. These results offer additional insights into the system's performance under different modality settings and further validate the contributions of multimodal integration to QSS tasks.

Table 10: Comparison of SDR values on AQSS among sound separation models trained with diverse modality data (**Text**, **Image** and **Audio**) and training methods (**MixUP**).

| Audio | Text | Image | MixUP | Mean SDR (AQSS) | Med SDR (AQSS) |
|---|---|---|---|---|---|
| ✓ | | | | $5.79 \pm 0.78$ | 5.19 |
| ✓ | ✓ | | | $6.67 \pm 0.71$ | 5.38 |
| ✓ | ✓ | ✓ | | $6.97 \pm 0.66$ | 5.40 |
| ✓ | ✓ | ✓ | ✓ | $7.12 \pm 0.65$ | 5.45 |

### C.5 IMAGEBIND ABLATION EXPERIMENTS

To evaluate the contribution of ImageBind pretraining to the performance of our model, we conducted ablation experiments focused on the role of ImageBind. As shown in Table **??**Our experiments indicate that without a pretrained model, performance drops significantly, as the model cannot effectively extract features or align representations across modalities. While fine-tuning ImageBind (E2) slightly improves in-domain performance, it hinders generalization to out-of-domain data, resulting in a 1.94 SDR drop on the MUSIC test set. Freezing ImageBind (E3) and using a linear layer provides a balance between performance and generalization.

Table 11: ImageBind Ablation Experiments.

| ID | Pretrained | Imagebind | VGGSOUND-CLEAN | | | MUSIC |
|---|---|---|---|---|---|---|
| | | | SDR(TQSS) | SDR(IQSS) | SDR(AQSS) | SDR(TQSS) |
| E1 | ✗ | tuning | 2.33 | 2.31 | 1.94 | - |
| E2 | ✓ | tuning | **6.81** | **6.73** | **7.22** | 4.76 |
| E3 | ✓ | **freeze** | 6.70 | 6.69 | 7.12 | **6.70** |

### C.6 ABLATION ON NEGATIVE QUERIES

When constructing the final query features with negative queries, it is crucial not only to ensure stability but also to remove the information corresponding to the negative query from the original query. If we use $(1 - \alpha)\mathbf{Q} + \alpha\mathbf{Q}_N$, both the negative query and the positive query content are retained, which contradicts the goal of using negative queries. On the other hand, directly subtracting ( $\mathbf{Q} - \alpha\mathbf{Q}_N$ ) can lead to instability and difficulty in choosing the coefficient $\alpha$, as discussed in

Section 4.2 and Figure 2 of the paper. As shown in Table 12, we provide the results for three methods with $\alpha = 0.5$ for a more intuitive comparison:

Table 12: Performance Comparison of Different Negative Query Embedding Strategies

| Query Embedding | SDR(TQSS) | SDR(IQSS) | SDR(AQSS) |
|---|---|---|---|
| $(1 - \alpha)\mathbf{Q} + \alpha\mathbf{Q}_N$ | 3.96 | 3.23 | 3.77 |
| $\mathbf{Q} - \alpha\mathbf{Q}_N$ | 6.77 | 6.92 | 6.65 |
| $(1 + \alpha)\mathbf{Q} - \alpha\mathbf{Q}_N$ | 7.57 | 7.68 | 7.22 |

## D  DATASETS

**VGGSOUND (Chen et al., 2020)**  VGGSOUND is a large-scale audio-visual dataset containing over 200,000 videos spanning 309 diverse sound categories, designed to support research in audio event classification, sound separation, and other multi-modal learning tasks.

**MUSIC (Zhao et al., 2018)**  The MUSIC dataset is a collection of 536 video recordings of people playing a musical instrument out of 11 instrument classes.

**Music-Clean+ (Dong et al., 2022)**  A refined subset of the music audio data, focusing on clean and distinct tracks with minimal noise or overlapping sounds. To prevent unintended overlaps of target sound types between the MUSIC and VGGSound datasets caused by label mismatches, all videos of musical instrument performances are excluded from the VGGSound dataset in this setting.

**VGGSound-Clean+ (Dong et al., 2022)**  A refined subset of the VGGSound dataset, comprising audio clips with reduced background noise and more distinct sound events. Dong et al. (2022) manually selects 100 clean samples containing clear and distinct target sounds from the VGGSound test set, referred to as VGGSound-Clean+.

## E  LIMITATIONS

This study still lacks more generalized validation. For the sake of comparison with prior models, we primarily conduct experiments on the VGGSOUND and MUSIC datasets. While these datasets are well-suited for omni-modal sound separation tasks, and VGGSOUND already encompasses over 300 sound events across various categories, they still fall short of representing all real-world audio events, particularly some abstract sound events like pop music, which are not adequately represented in the dataset. Moving forward, we aim to construct a more extensive dataset (Cheng et al., 2024) to address this limitation. We also plan to use RLHF (Han et al., 2024b;a; Kumar et al., 2024) to optimize sound separation and improve its generalization.

## F  ETHICAL DISCUSSION

The sound separation method proposed in this paper enables the separation of target queries according to user preferences, but it may be susceptible to misuse, such as separating background music (BGM) and vocals from copyrighted songs, extracting dialogue and various sound effects from copyrighted movies, and so on. However, given the long history of development in this field and the relatively clear copyright ownership of various audio and video products, the potential social impact is not expected to be significant.

