# OpenReview forum: "OmniSep: Unified Omni-Modality Sound Separation with Query-Mixup"
_ICLR.cc/2025/Conference — ICLR 2025 Poster_

### Official Review · Reviewer_nQk6 · 2024-10-28

**Soundness:** 3
**Presentation:** 3
**Contribution:** 3
**Rating:** 6
**Confidence:** 4

**Summary:**

This paper attempts to cover text-, image- and audio-queried sound separation all at once with one model by exploiting ImageBind as its encoder. Several training techniques are introduced and verified its effectiveness in terms of signal-to-noise ratio as well as spectral similarity, which demonstrates the superiority of OminiSep to other existing query-based sound separation methods

**Strengths:**

- Three novel training techniques, Query-mixup, NQ, Query-Aug, are proposed and evaluated in the standardized experimental settings
- Consistent improvements of median SDR in Table 2, which demonstrates the SOTA performance of OmniSep in the query-based sound separation field
- Good visualization of ImageBind embeddings to show the clear motivation of the Query-mixup

**Weaknesses:**

- Inappropriate motivation: I felt puzzled when I was reading the introduction part because the original motivation of the paper was how we could increase the number of high-quality training dataset although this motivation was never addressed anywhere in the other sections of the paper. If the objective is like the above, why don't we simply compare your scheme to other data augmentation methods or artificial data creation methods other than sound separation? I agree that denoising with sound separation would serve as one of many methods to achieve the same goal. Then, I believe the author should prove that OmniSep is the one of the best methods to increase the number of data. That being said, while assessing the paper, I somehow felt that this was not your intention. Since the technical contributions of the paper are fair enough, I suggest rewriting the introduction part in a way like your motivation is to improve the quality or coverage of text-, image-, or audio-queried sound separation and NOT to use sound separation to increase the number of training datasets for other purposes.
- As an expert in this field, I don't agree relying only on SDR to measure the performance of any sound separation methods. Rather than showing pairs of spectrums, I suggest conducting a listening test to doublecheck if the performance of OmniSep really dominates others and the difference is recognizable. If the goal of the paper is not to make separated data available to human, as you originally stated in the introduction part, i.e., data increase is the objective, I don't think having a subjective listening test enriches the content. Otherwise, I strongly recommend this because this is a common practice in the sound separation community.

**Questions:**

The motivation of Query-mixup is clear. On the other hand, its effectiveness compared to finetuning of ImageBind is not clear. Why don't you simply finetune your encoder so that all the modalities align well each other to the inputs of your interests. It is also unclear if the performance improvement really comes from the proposed three methods because the encoders in CLIPSep and OmniSep are different. Table 3 is a nice way to prove Query-Aug is actually important to increase SDR. Not sure about other two. You could have tried the rest of two methods on CLIPSep and see the performance improvements of CLIPSep?

---

> ### Author Response · Authors · 2024-11-24
> **Response to Reviewer nQk6**
>
> Thank you for recognizing the contributions of our work, including the effectiveness of OmniSep and the rationality of the ImageBind embedding visualization analysis. Please allow me to address your questions in detail:
>
> **Q1: Original Motivation for OmniSep**
>
> **A1**: Thank you for your suggestions regarding our paper. Our intention was to highlight the significant value sound separation could bring to the research community. Based on your feedback, we realized that our initial framing might have limited the perceived scope of sound separation’s potential impact. In the latest version, we have rewritten the introduction to better emphasize the importance of cross-modal query sound separation, particularly focusing on homologous but heterogeneous modalities.
>
> **Q2: Subjective Testing**
>
> **A2**: We conducted Mean Opinion Score (MOS) evaluations to compare the sound separation results of several major models. The results are as follows:
>
> |  | MOS |
> | --- | --- |
> | CLIPSEP(T) | 3.36 |
> | CLIPSEP(I) | 3.51 |
> | AudioSep(T) | 3.85 |
> | OmniSep(A) | 3.94 |
> | OmniSep(I) | 3.83 |
> | OmniSep(T) | 3.89 |
> | OmniSep(I+A+T) | 4.01 |
> | OmniSep(I+A+T)+negative query | 4.14 |
> | Real audio | 4.32 |
>
> These experimental results have been updated in Table 9 in Appendix C.3 of the latest version.
>
> **Q3: ImageBind Fine-Tuning vs. OmniSep**
>
> **A3**: In prior works, researchers have found that models like CLIP and ImageBind already achieve well-aligned representations, enabling effective cross-modal mapping with a simple linear layer. In this work, the key focus of Query-Mixup is to enable the model to process inputs from three modalities simultaneously, ensuring that performance across modalities remains unaffected.
>
> To compare, we evaluated the performance of fine-tuning ImageBind combined with the iterative training strategy used in CLIPSEP against training OmniSep with Query-Mixup. The results are shown below. As observed in experiments E1 and E2, fine-tuning ImageBind yields only marginal improvements, with performance close to that of linear fine-tuning. However, when Query-Mixup is used, the model (E3) achieves significant performance gains across all single-modality tasks, highlighting the effectiveness of Query-Mixup.
>
> |  | imagebind tuning | training strategy | Mean SDR(TQSS) | Mean SDR(IQSS) | Mean SDR(AQSS) |
> | --- | --- | --- | --- | --- | --- |
> | E1 | ✘ | iterative modality training | 6.37 | 6.53 | 6.97 |
> | E2 | ✔️ | iterative modality training | 6.40 | 6.52 | 6.93 |
> | E3 | ✘ | query-mixup | **6.70** | **6.69** | **7.12** |
>
> **Q4: Ablation Study on Query-Mixup and Negative Query for CLIPSEP**
>
> **A4:** Following your suggestion, we conducted an ablation study on VGGSOUND-Clean to evaluate the impact of Query-Mixup and negative query methods on CLIPSEP. The results are as follows:
>
> |  | Method | Mean SDR（TQSS） | Mean SDR（IQSS） |
> | --- | --- | --- | --- |
> | E4 | CLIPSEP | 5.49 | 5.46 |
> | E5 | CLIPSEP+Query-mixup | 5.67 | 5.74 |
> | E6 | CLIPSEP+negative query | **6.32** | **6.17** |
>
> Furthermore, with the help of Query-Mixup, our model is the first to achieve sound separation using fully composed queries from all modalities. As shown in Table 1, this approach significantly improves performance for Composed Omni-Modal Queried Sound Separation by leveraging homologous but heterogeneous information from multiple modalities, setting a new benchmark in the field.
>
> |  | Query modality | Mean SDR |
> | --- | --- | --- |
> | OmniSep | T | 6.70 |
> | OmniSep | T+I | 7.12 |
> | OmniSep | T+I+A | **7.46** |
>
> We hope this clarifies your questions and demonstrates the strength of our proposed methods. Please feel free to reach out with additional feedback or inquiries!

---

> > ### Comment · Reviewer_nQk6 · 2024-11-25
> >
> > Thanks for your response and efforts to address all my concerns. Now the story looks more natural. I'm happy to raise the score to 6

---

> ### Author Response · Authors · 2024-11-26
> **Appreciation to Reviewer nQk6 for Valuable Suggestions and Positive Comments**
>
> Thank you for your thoughtful feedback, which has greatly enhanced the clarity and quality of our paper. The revised version now feels much more natural, and we sincerely appreciate your valuable input. If you have any additional questions or suggestions, please do not hesitate to reach out.

---

### Official Review · Reviewer_Mat5 · 2024-10-30

**Soundness:** 3
**Presentation:** 2
**Contribution:** 2
**Rating:** 6
**Confidence:** 3

**Summary:**

This paper proposes an audio source separation model that can be queried via audio, language and image. To construct such model, this paper proposes query mixup which blends query from different modalities during training and ensure queries from different modalities reside in the same semantic space. This paper also propose a negative query mechanism to enhance the flexibility of query in inference stage. Last, to enable open-vocabulary text query input to the model which trained on close-set class name, this paper propose a query-augmentation method at inference time which retrieves nearest class name. Experiment result shows the model outperforms existing model in all modalities on MUSIC, VGGSOUND-CLEAN+, and MUSIC-CLEAN+ datasets.

**Strengths:**

1. A novel model for source separation queried by audio, text and image. Propose a novel query-mixup method to enable such model.
2. Propose novel negative query and query-aug method to improve performance and flexibility at inference time.
3. State-of-the-art performance on separation tasks queried by all modalities

**Weaknesses:**

1. Presentation: Certain aspects of the paper’s presentation feel oversimplified, particularly in explaining key contributions and technical details. Please refer to the questions section for specific areas needing clarification.

2. Contribution of Query-Aug: The significance of the query-augmentation (query-aug) contribution seems overstated. For example, lines 054–056 suggest that current systems cannot handle open-vocabulary queries. However, Liu et al. (2023) ("Separate Anything You Describe") demonstrates that open-vocabulary language queries are achievable using audio-text datasets, such as Clotho or AudioCaps. Furthermore, the model proposed in this paper should be capable of training on these audio-text datasets and potentially on datasets lacking complete audio, text, or image pairs. An expanded discussion comparing this work to Liu et al. (2023), along with a rationale for the choice of datasets, would strengthen the contribution.

3. Experimental Results: The presentation of experimental results lacks some necessary detail:
 - What training settings were used for the models in Table 2, and how do these models (particularly model #5) correspond to those in Table 1?
 - For Table 4, a comparison of results from querying the same audio mixture across different modalities would provide valuable insights.

4. Metrics for Comparison: In prior works cited in the paper, metrics such as SI-SDR and SDRi are commonly used for source separation tasks, as they better reflect model performance in recent studies. Including these metrics or providing justification for the metrics chosen would enhance the comparative analysis.

**Questions:**

The introduction of the Separate-Net is missing some details:
1. Line 210: what is k and what does k corresponds to?
2. Does the query conditioned to the U-Net? If not, why query is not conditioned to UNet? If possible, it would be helpful to visualize the qi in some examples. It feels odd to me that the query only results in a channel-wise weight.

---

> ### Author Response · Authors · 2024-11-24
> **Response to Reviewer Mat5**
>
> Thank you for recognizing the flexibility and performance of our work. Please allow us to clarify your questions in detail:
>
> **Q1: Applicability of Query-Aug**
>
> **A1**: Natural sounds can be described using either class labels or textual descriptions. Class labels provide coarse-grained descriptions, while textual descriptions capture fine-grained details, emphasizing subtle differences between sounds. While AudioSep [1] achieves open-vocabulary sound separation using caption-based descriptions, its performance on unseen queries is still unstable due to data limitations. By combining AudioSep with our proposed Query-Aug method, we demonstrate that Query-Aug further enhances model performance, making open-vocabulary queries more robust. Related experimental results can also be found in Section D of the demo page.
>
> | Method | Training Data (hours) | Mean SDR | Med SDR |
> | --- | --- | --- | --- |
> | CLIPSEP-text | 550 | 3.53±0.52 | 2.91 |
> | CLIPSEP-text+Query-Aug | 550 | **5.24±0.79** | **4.87** |
> | AudioSep | 14,000 | 7.24±0.67 | 6.23 |
> | AudioSep+Query-Aug | 14,000 | **7.46±0.63** | **6.34** |
>
> **Q2: Reason for Dataset Selection**
>
> **A2**: Considering that the key contribution of this work is the proposal of an omni-modality sound separation model, we selected datasets that support diverse modalities. In prior vision-queried sound separation tasks, most works have been conducted on VGGSOUND. To enable fair comparisons, we followed CLIPSEP’s experimental setup. However, your suggestion is highly valuable. Based on your feedback, we expanded our experiments by incorporating additional data from AudioSet into VGGSOUND, leading to further performance improvements. Here is the futher experiments:
>
> ***Composed Omni-Modal Queried Sound Separation***
>
> |  | Training Data | VGGSOUND-Clean+ |
> | --- | --- | --- |
> | OmniSep | VGGSOUND | 7.46±0.65 |
> | OmniSep | AudioSet+VGGSOUND | **7.63±0.62** |
>
> **Q3: Results of the Same Query Across Different Modalities**
>
> **A3**: In Sections A.1, A.2, and A.3 of the demo page, we present the results of using the same query across different modalities. You are welcome to review these sections for detailed insights.
>
> **Q4: SISDR and SDRi Metrics**
>
> **A4**: Thank you for your suggestion! Based on your feedback, we added SI-SDR and SDRi metrics to our evaluation. However, since the CLIPSEP paper only provided partial checkpoints for reference models, we compare performance with all available baselines. These experimental results have been updated in Table 9 in Appendix C.3 of the latest version.
>
> |  | SI-SDR | SDRi |
> | --- | --- | --- |
> | CLIPSEP | 3.92 | 5.32 |
> | CLIPSEP(I) | 4.32 | 5.27 |
> | AudioSep | 5.43 | 5.94 |
> | OmniSep(T) | 5.53 | 6.64 |
> | OmniSep(I) | 5.49 | 6.68 |
> | OmniSep(A) | 6.12 | 7.08 |
> | OmniSep(T+I+A) | 6.56 | 7.37 |
>
> **Q5: Experimental Settings in Table 2**
>
> **A5**: All experiments in Table 2 follow the same settings as the OmniSep experiments in Table 1, with changes only to the training modalities and training strategy (Query-Mixup). Detailed settings are described in Appendix A. Experiment #5 in Table 2 corresponds to text-queried and image-queried sound separation experiments for OmniSep (ours) in Table 1. Other experiments represent ablation studies and do not correspond directly to experiments in Table 1.
>
> **Q6: Separate-Net Experimental Details**
>
> **A6**: For a fair comparison, our experiments were based on the CLIPSEP framework. The parameter  $k$  is a hyperparameter set to 32, following the setup in CLIPSEP.
>
> To ensure comparability with the benchmark established by CLIPSEP, we injected features into the output of the final layer of the UNet model (in the form of mask weights q_i), as used in CLIPSEP [2] and SOP [3].
>
> To further clarify differences between injection methods, we conducted a theoretical analysis:
>
> 1. Feature Injection Method in AudioSep: Features are injected into hidden embeddings at every layer, enabling multi-level control over sound separation embeddings.
> 2. Feature Injection Method in CLIPSEP: Features are injected directly into the final layer, providing more direct control over the output mask.
>
> Despite these differences, both architectures achieve the goal of sound separation. In our experiments, AudioSep’s approach yielded better results, as shown in Table 1, which compares CLIPSEP and AudioSep. However, to maintain experimental consistency, we adopted the current Separate-Net structure in this version of the paper. However, we are committed to including an OmniSep implementation based on the AudioSep architecture in our open-sourced code.
>
> We hope this clarifies your questions. Please feel free to reach out with further feedback or inquiries!
>
> [1] Liu X, Kong Q, Zhao Y, et al. Separate anything you describe[J]. arXiv 2023.
>
> [2] Dong H W, Takahashi N, Mitsufuji Y, et al. Clipsep: Learning text-queried sound separation with noisy unlabeled videos[J]. ICLR2022
>
> [3] Zhao H, Gan C, Rouditchenko A, et al. The sound of pixels[C]. ECCV2018

---

> > ### Comment · Reviewer_Mat5 · 2024-11-25
> >
> > Thank you to the authors for providing a detailed response and updating the experiment results. Your clarifications have addressed my concerns effectively. As a result, I have updated my score to 6.

---

> > > ### Author Response · Authors · 2024-11-25
> > >
> > > Thank you for your thoughtful review and constructive suggestions. I am delighted to know that all your questions have been addressed. Your valuable feedback has greatly contributed to enhancing the quality of our paper.

---

### Official Review · Reviewer_PPUR · 2024-11-02

**Soundness:** 3
**Presentation:** 1
**Contribution:** 2
**Rating:** 6
**Confidence:** 3

**Summary:**

This paper presents OmniSep, a novel unified framework for query-based sound separation that accommodates multiple query modalities (text, image, and audio) within a single model. The authors introduce three key technical contributions: Query-Mixup for simultaneous multi-modal query processing, a negative query mechanism for unwanted sound suppression, and Query-Aug for handling natural language descriptions beyond predefined class labels. The model's architecture represents an advance over previous approaches, which were typically constrained to single-modality queries.

Experimental validation on several datasets demonstrates OmniSep's performance compared to existing methods across all query modalities. The model exhibits robust separation capabilities in complex, multi-source scenarios and achieves state-of-the-art results on MUSIC, VGGSOUND-CLEAN+, and MUSIC-CLEAN+ benchmarks. OmniSep's multi-modal query capability enables enhanced separation performance through the simultaneous application of different query types, such as combining textual and visual queries.

**Strengths:**

The paper presents a clear and well-motivated problem statement, addressing three fundamental limitations in current sound separation approaches: the absence of unified multi-modal query handling, insufficient flexibility in sound manipulation (particularly for unwanted sound removal), and restricted vocabulary constraints that preclude natural language descriptions. The authors construct a compelling narrative throughout the introduction and literature review, effectively contextualizing their contributions within the field through comprehensive citations and thorough analysis of related work.

The methodology is generally well-documented to begin with. The experimental validation is particularly robust, encompassing diverse tasks and datasets that demonstrate the method's versatility. The authors provide extensive ablation studies that illuminate the model's internal mechanisms and justify some of the architectural choices. Their commitment to reproducibility through code release further strengthens the paper's contribution to the field.

**Weaknesses:**

There are several issues with the paper, which can broadly be classified into two areas.  Most of these issues can be fixed with better, scientific writing, and more explanation, rigour and experimentation.

Technical Clarity Issues:
1. Significant documentation gaps in core variables and operations in Separate-Net:
   - q_i and q_ij are poorly explained or completely undefined in Separate-Net
   - Many variables are undefined and or with no dimensions specified. All variables should be clearly defined with dimensions given.
   - M(hat) is defined but never used.
   - Mechanism of how masks are used/applied is not explained.
   - Audio U-Net lacks both citation and architectural explanation.
2. Query-Aug undefined components:
   - Q_des: completely undefined without dimensions and no explanation of how these features are obtained (T5/Bert?).
   - Query-Set: is undefined and dimensions not specified.
   - Q_aug defined as argmax but its integration into model never explained.
   - sim(.,.) is not a standard operator, it should be properly defined (cosine similarity?).

Experimental Validation Problems:
1. Table 1 uses a 2017 model called PIT for non-queried sound separation. State-of-the-art audio-only separation methods (i.e. TDANet, TF-GridNet) would serve stronger baselines (while these models are for speech separation, so is PIT. Additionally, PIT was proposed so solve the permutation problem, not as a strong speech separation model).
2. Table 2 does not show that the query-mixup method works, only that it scales across different modalities. Validating its effectiveness would mean comparing to other methods and/or different strategies.

However, there are some other issues. While the method is interesting, the paper's novel work can be summarised as a weighted average of modalities, a linear layer and (1+alpha)Q-alphaQ_N for negative queries.

**Questions:**

1. Separate-Net only converts to magnitude spectrum X, ignoring phase. Modern time-frequency methods (RTFS-Net, TF-GridNet) use retain both magnitude and phase information via concatenated real/imaginary STFT outputs. Some methods additionally concatenate the magnitude to create a three channel representation. Have you tried these approaches?

2. Why keep ImageBind parameters frozen? What if it was trained from scratch with the model? What if it was initialized and fine tuned with the model using a lower learning rate?

3. For the Negative Query, why (1+alpha)Q-alphaQ_N instead of (1-alpha)Q+alphaQ_N, Q-alphaQ_N or a scale-and-shift approach such as in stable diffusion - please add some justification in this section or some experimental evidence.

4. And the problems raised in the weaknesses section.

---

> ### Author Response · Authors · 2024-11-24
> **Response to Reviewer PPUR (1/N)**
>
> Thank you for recognizing the significance of our work and the sufficiency of our experiments. Please allow me to address your questions in detail:
>
> **Q1: Clarity of the Technique**
>
> **A1**: Your meticulous reading is truly appreciated. Thanks to your thoughtful suggestions, we have added the details you mentioned in the latest version of the paper. We believe these improvements greatly enhance the presentation of our work.
>
> **Q2: Non-queried Sound Separation**
>
> **A2**: The PIT experimental results were derived from the baseline constructed by CLIPSEP[1]. Specifically, it is the prior research [2], which used PIT for sound separation. Also noted in CLIPSEP that:
>
> > The PIT model requires a post-selection step to get the correct source. Without the post-selection step, the PIT model returns the right source in only 50% of cases.
> >
>
> We conducted similar tests using TDANET [3] and have included the results in the latest version of the paper to provide a more comprehensive comparison.
>
> **Q3: Effectiveness of Query-Mixup**
>
> **A3:** Among existing multi-modal query sound separation approaches [1,4], there are two main strategies: iterative modality training, as exemplified by CLIPSEP, and our proposed Query-Mixup approach. We present an ablation study on the query embedding operation strategies, demonstrating that our Query-Mixup strategy consistently improves performance across tasks:
>
> |  | training strategy | Mean SDR(TQSS) | Mean SDR(IQSS) | Mean SDR(AQSS) |
> | --- | --- | --- | --- | --- |
> | E31 | iterative modality training | 6.37 | 6.53 | 6.97 |
> | E32 | query-mixup | **6.70** | **6.69** | **7.12** |
>
> **Q4: Discussion on Phase-Based Methods**
>
> **A4**: Indeed, these two works [5,6] have achieved state-of-the-art performance in speech separation. I have read these studies before and am impressed by their remarkable performance in terms of both accuracy and model scale. In this work, however, our primary focus is on investigating multi-modal collaboration between different query modalities and the effect of composed queries from omni modalities on sound separation performance. Therefore, phase information was not included in this study. Nevertheless, I am eager to explore similar methods in our future work.

---

> > ### Author Response · Authors · 2024-11-24
> > **Response to Reviewer PPUR (2/N, N=2)**
> >
> > **Q5: Use of ImageBind**
> >
> > **A5**: In many existing works [7], researchers have observed that representations in models like CLIP and ImageBind are already well-aligned, allowing a simple linear mapping layer to suffice for cross-modal mapping. In this work, the focus of Query-Mixup is to enable the model to process inputs from three modalities simultaneously without mutual interference. For comparison, we conducted the following experiments:
> >
> > - E50: Results from a randomly initialized model trained with a learning rate of 1e-5.
> > - E51: Results from fine-tuning a pretrained ImageBind model.
> > - E52: Results from freezing ImageBind and fine-tuning only a linear mapping layer.
> >
> > Our experiments indicate that without a pretrained model, performance drops significantly, as the model cannot effectively extract features or align representations across modalities. While fine-tuning ImageBind (E52) slightly improves in-domain performance, it hinders generalization to out-of-domain data, resulting in a 1.94 SDR drop on the MUSIC test set. Freezing ImageBind (E53) and using a linear layer provides a balance between performance and generalization.
> >
> > ***Experiments on VGGSOUND-clean***
> >
> > |  | pretrained | imagebind  | Mean SDR(TQSS) | Mean SDR(IQSS) | Mean SDR(AQSS) |
> > | --- | --- | --- | --- | --- | --- |
> > | E50 | ✘ | tuning | 2.33 | 2.31 | 1.94 |
> > | E51 | ✔️ | tuning | 6.81 | 6.73 | 7.22 |
> > | E52 | ✔️ | freeze | **6.70** | **6.69** | **7.12** |
> >
> > ***Experiments on MUSIC***
> >
> > |  | pretrained | imagebind  | Mean SDR(TQSS on MUSIC) |
> > | --- | --- | --- | --- |
> > | E51 | ✔️ | tuning | 4.76 |
> > | E52 | ✔️ | freeze | **6.70** |
> >
> > **Q6: Discussion on Negative Queries**
> >
> > **A6**: When constructing the final query features with negative queries, it is crucial not only to ensure stability but also to remove the information corresponding to the negative query from the original query. If we use  $(1-\alpha)\mathbf{Q} + \alpha\mathbf{Q}_N$ , both the negative query and the positive query content are retained, which contradicts the goal of using negative queries. On the other hand, directly subtracting ( $\mathbf{Q} - \alpha\mathbf{Q}_N$ ) can lead to instability and difficulty in choosing the coefficient \alpha, as discussed in Section 4.2 and Figure 2 of the paper. Below, we provide the results for three methods with $\alpha=0.5$ for a more intuitive comparison:
> >
> > |  | query embedding | TQSS | IQSS | AQSS |
> > | --- | --- | --- | --- | --- |
> > | E61 | $(1-\alpha)\mathbf{Q}+\alpha \mathbf{Q}_N$ | 3.96 | 3.23 | 3.77 |
> > | E62 | $\mathbf{Q}-\alpha \mathbf{Q}_N$ | 6.77 | 6.92 | 6.65 |
> > | E63 | $(1+\alpha)\mathbf{Q}-\alpha \mathbf{Q}_N$ | **7.57** | **7.68** | **7.22** |
> >
> > We hope this addresses all your questions clearly! Please feel free to reach out with further queries.
> >
> > [1] Dong H W, Takahashi N, Mitsufuji Y, et al. Clipsep: Learning text-queried sound separation with noisy unlabeled videos[J]. ICLR2022
> >
> > [2] Kavalerov I, Wisdom S, Erdogan H, et al. Universal sound separation[C] ICASSP2019
> >
> > [3] Li K, Yang R, Hu X. An efficient encoder-decoder architecture with top-down attention for speech separation[J]. ICLR2023
> >
> > [4] Liu X, Kong Q, Zhao Y, et al. Separate anything you describe[J]. arXiv 2023.
> >
> > [5] Pegg S, Li K, Hu X. RTFS-Net: Recurrent time-frequency modelling for efficient audio-visual speech separation[J]. arXiv preprint ICLR2024
> >
> > [6] Wang Z Q, Cornell S, Choi S, et al. TF-GridNet: Making time-frequency domain models great again for monaural speaker separation[C]//ICASSP 2023-2023 IEEE International Conference on Acoustics, Speech and Signal Processing ICASSP2023
> >
> > [7] Wang Z, Zhang Z, Cheng X, et al. FreeBind: Free Lunch in Unified Multimodal Space via Knowledge Fusion[C]. ICML2024

---

> > > ### Comment · Reviewer_PPUR · 2024-11-25
> > >
> > > Thank you for providing such detailed experiments in such a short period of time, it's extremely impressive.
> > >
> > > Would it be possible to add these experiments to the manuscript and reference them at the pertinent parts in the text (i.e. we use this method because 'x', see appendix 'y'). The same goes for the experiments requested by other reviewers. They provide important context and make the work much more thorough.
> > >
> > > A few dimensions are still unlabelled. Having the dimensions readily available makes reading a much nicer experience, and makes the manuscript more comprehensive:
> > >  - $Q_i$ a slice in the time dimension of $Q$, i.e. $R^{1024}$?
> > >  - $\tilde{M}$ is a set of $k$ elements, but what are the dimensions of each element?
> > >  - $i$ is unclear. Should the number of predicted masks $\hat{M}$ be equal to the number of intermediate masks ($k$)? Is $i \in \{0,...,n-1\}$?
> > >  - Define the dimensions of the $\hat{M}_i$ and $\hat{M}$.
> > >  - The $i$ and $j$ notation is difficult to follow. $i$ is the number of time steps, hence $i \in \{0,...,n-1\}$, and then we have $i$ masks. So we have 1 mask for each time step. But it is chosen that $k\leq n$. But $k$ is the output dimension of a linear layer, meaning the input dimensions needs to be defined. Are you using a linear layer to compress the time dimensions and if so, does that mean the input audio size has to be fixed ahead of time i.e. the method does not generalize to any length of audio?
> > > - How are the $Q_i$ obtained from $A_i, V_i, T_i$, and what are the dimensions of $A_i, V_i, T_i$? How do you ensure that audio features, video features and text features all have the same number of time steps? Surely the video features would have much fewer time steps for 25 fps video, etc.

---

> > > > ### Author Response · Authors · 2024-11-25
> > > > **Futher Response to Reviewer PPUR**
> > > >
> > > > Thank you for your follow-up response and for recognizing our efforts during the rebuttal phase. Your feedback has helped us identify that our description in this section may still be prone to misinterpretation. Please allow us to provide further clarification.
> > > >
> > > > **Q1: Definition of $i$**
> > > >
> > > > **A1**: For a mixed audio signal  $A_{\text{mix}}$  composed of  $n$  audio sources  {$\{A_1, A_2, \cdots, A_n\}$} , each audio source  $A_i$  is associated with a corresponding video  $V_i$  and textual query  $T_i$ , forming  $n$  triplets  {$\{(A_1, V_1, T_1), \cdots, (A_n, V_n, T_n)\}$}. The query  $\mathbf{Q}_i$  for sound separation is derived from the  $i$ -th triplet  $(A_i, V_i, T_i)$ . The predicted mask  $\hat{M}_i$  corresponds to the audio source  $A_i$ .
> > > >
> > > > **Q2: Definition of $j$**
> > > >
> > > > A2: The spectrogram  X  is fed into the audio U-Net to obtain $k$ intermediate masks $\tilde{M}=${$\{\tilde{M}_1, \cdots, \tilde{M}_j, \cdots, \tilde{M}_k\}$}
> > > >
> > > > , where $\tilde{M} \in \mathbb{R}^{k \times F \times T}$, $\tilde{M}_{j}$ is the $j$-th intermediate mask.
> > > >
> > > > **Q3: Dimensions of  $Q_i$  and alignment of audio, visual, and text queries**
> > > >
> > > > **A3**: ImageBind extracts global semantic representations for different modalities. Unlike representations commonly used in speech separation tasks, such as AV-HuBERT or HuBERT, ImageBind’s embeddings do not contain temporal dimension. As mentioned on line 312, for video queries, we sample four frames at 1-second intervals and average their embeddings to obtain the image embedding  $Q_v \in \mathbb{R}^{1024}$ . For audio and text queries, ImageBind extracts the entire segment into an audio embedding  $Q_a \in \mathbb{R}^{1024}$  and a text embedding  $Q_t \in \mathbb{R}^{1024}$ , respectively. After the weighted combination in Equation (1),  $Q_i$  has the same dimension as the query embeddings for each modality,  $Q_i \in \mathbb{R}^{1024}$ .
> > > >
> > > > **Q4: Dimensions of**  $\hat{M}_i$ and $\tilde{M}$
> > > >
> > > > **A4**: The mixed audio is first converted into the magnitude spectrum  $X \in \mathbb{R}^{C \times F \times T}$  using the Short-Time Fourier Transform (STFT), where  $C$  represents the number of channels,  $F$  is the frequency dimension, and  $T$  is the time dimension. For single-channel audio in this paper,  $C$ = 1 . Using the audio U-Net, we extract  $\tilde{M}$ , which contains  $k$  intermediate masks,  $\tilde{M} \in \mathbb{R}^{k \times F \times T}$ . The query embedding  $Q_i$  is passed through a linear layer to compute weights for the  $k$  intermediate masks, which are then combined with  $\tilde{M}$  to produce the predicted mask corresponding to  $A_i$ ,  $\hat{M}_i \in \mathbb{R}^{C \times F \times T}$ . Since the audio U-Net allows the time dimension  $T$  to be of arbitrary length, our model can handle audio signals of any length.
> > > >
> > > > **We have updated this section in the latest version of the paper and look forward to your further feedback.** We hope the revisions in this version clarify the remaining ambiguities.
> > > >
> > > > ---
> > > >
> > > > Additionally, regarding the extra experimental results, while we have incorporated some of the reviewer-suggested experiments into the paper, there are still a few that have not yet been fully integrated. We are actively working to include these additional experiments in the final version. We will ensure that all updates are completed before the deadline. To provide a prompt response, the latest version update only includes revisions addressing these specific misunderstandings (lines 200–211).

---

> > > > ### Author Response · Authors · 2024-11-27
> > > > **Latest Paper Version Uploaded**
> > > >
> > > > We have updated the latest version of our paper, incorporating all the experiments conducted during the rebuttal period into the appendix. Additionally, we have addressed the definitions of parameters such as i and j, as highlighted in your previous feedback. We look forward to receiving your further comments and hope that this version resolves any potential misunderstandings.
> > > >
> > > > Thank you again for your valuable feedback!

---

> > > > > ### Author Response · Authors · 2024-12-01
> > > > > **Follow-Up on Reviewer PPUR’s Comments**
> > > > >
> > > > > Dear Reviewer PPUR,
> > > > >
> > > > > We have further revised our paper to address your concerns and have included the corresponding experimental results. As the rebuttal phase is drawing to a close, we kindly request your feedback at your earliest convenience. We hope our revisions have satisfactorily addressed all your questions.
> > > > >
> > > > > Best regards,
> > > > > Authors

---

> > > > > > ### Comment · Reviewer_PPUR · 2024-12-02
> > > > > >
> > > > > > Thank you for your hard and diligent work on updating the paper to include more explanations and experiments. The methods are much clearer now, and the new experimental results add depth and robustness to the study. Updating the score to 6.

---

### Official Review · Reviewer_7ccM · 2024-11-04

**Soundness:** 3
**Presentation:** 3
**Contribution:** 3
**Rating:** 6
**Confidence:** 3

**Summary:**

This paper presents OmniSep, a framework for omni-modal sound separation, which supports sound isolation using queries from multiple modalities, such as text, image, and audio, either independently or in combination. The key method of omni-modal source separation is the introduction of Query-Mixup, a strategy that mixes query features from different modalities, using a pre-trained ImageBind. OmniSep further enables open-vocabulary sound separation with Query-Aug, a retrieval-augmented method that enhances adaptability, particularly for text-based queries. Experimental evaluations on MUSIC, VGGSOUND-CLEAN+, and MUSIC-CLEAN+ datasets showcase OmniSep’s SOTA performance across various modality-based separation tasks.

**Strengths:**

The paper presents an omni-modal sound separation approach, allowing users to conduct sound separation tasks using queries across various modalities, including text, image, and audio, both independently and jointly. Additionally, the authors conducted a comprehensive evaluation across several benchmarks. Further, the inclusion of extensive ablation studies provides insight into the contributions of each component: impact of Query-Mixup, negative query weighting, long text description-queried separation, and analysis on query embeddings.

**Weaknesses:**

While the paper presents a solid framework, there are some details missing that make it difficult to give a firm acceptance (please answer Questions).

The use of full-length video as a query, which includes the target segment, raises questions about the potential for information leakage. It’s unclear if the QueryNet architecture ensures a bottleneck to prevent the model from “cheating” by directly accessing target audio features. To better demonstrate the model’s robustness, an alternative setup might involve using a different video as the query that shares the same class identity as the input audio.

Additionally, the comparison in Section 4.3 between the proposed weighting and a naive weighting approach on $Q_N$ may not be particularly insightful, as both methods use the same model. The proposed approach always has a weight difference of 1 ($1+\alpha$ vs. $\alpha$), while the comparison (naive) approach reduces the weight gap between $Q$ and $Q_N$, hence the results in Figure 2 are somewhat predictable. However, the insights provided by varying $\alpha$ are valuable.

**Questions:**

- What happens when the query is some class that’s not present inside the input mixture audio?
- Table 2: why no results on AQSS?
- Query-Aug Adaptability:
    - Is the Query-Aug method adaptable only to text queries $Q_T$, or does it extend to other modalities as well?
    - Additionally, how will the model handle text prompts containing multiple sound sources (e.g., “the sound of a baby with her parent’s soothing voice”)? Is the model trained to handle multi-source separation?
- Number of Audio Sources in Mixtures: For AQSS VGGSOUND, does $\mathcal{S}=5$ mean $A_\text{mix}=\sum_{n=1}^6A_n$? then for all other setups, is $A_\text{mix}=A_1+A_2$?
- Did the authors use the pre-trained ImageBind weights from the official repository (https://github.com/facebookresearch/ImageBind), or did they train ImageBind from scratch? They should mention the repository if pre-trained weights were used. If trained from scratch, they should include the details of the pre-training specifications.
- How does OmniSep respond when a query references a class that is not present within the input audio mixture?
- Why are there no AQSS results reported in Table 2?

Minor comments
- typo @Figure 1: “…, donated as $\text{Q}_T$, …”
- Figure 1: inside the Query-Mixup block, the weight factors are denoted as $W$, where it should be $w$
- grammar @line 217: “The training …”
- Table 1: should note the source of VGGSOUND-CLEAN+ and MUSIC-CLEAN+ subsets. Also provide a couple of sentences for details of these subsets.

---

> ### Author Response · Authors · 2024-11-24
> **Response to Reviewer 7ccM**
>
> Thank you for recognizing our framework and ablation experiments! Below are our detailed responses to your questions:
>
> **Q1: In AQSS, is the raw audio used as a query? What does $S=5$ mean in AQSS?**
>
> **A1**: In the AQSS setup of this paper, we use audio samples that shares the same class identity as the target audio as queries. We apologize for any confusion caused by insufficient description. This approach ensures that no audio information leakage occurs during the experiments. During inference, for VGGSOUND, we select 5 audio features from the category of the target audio and use their averaged feature as the audio query feature for that category.
>
> **Q2: How does OmniSep respond to a query that does not exist in the audio?**
>
> **A2:** When the query corresponds to content not present in the audio, the model is supposed to output silence.
>
> **Q3: Results for AQSS in Table 2?**
>
> **A3:** Due to space limitations, this part of the experiment was not included in the previous version of the paper. However, we believe the experiments on TQSS and IQSS already sufficiently support the conclusions of the section. Here, we provide the detailed results for AQSS:
>
> | Audio | Text | Image | MixUP | Mean SDR(AQSS) | Med SDR(AQSS) |
> | --- | --- | --- | --- | --- | --- |
> | ✔️ |  |  |  | 5.79±0.78 | 5.19 |
> | ✔️ | ✔️ |  |  | 6.67±0.71 | 5.38 |
> | ✔️ | ✔️ | ✔️ |  | 6.97±0.66 | 5.40 |
> | ✔️ | ✔️ | ✔️ | ✔️ | **7.12±0.65** | **5.45** |
>
> **Q4: Query-Aug adaptability?**
>
> **A4:** Yes, Query-Aug can be adapted to any modality, even modalities that have never been trained, such as 3D. For scenarios with multiple sound sources, one feasible approach is to use Query-Aug to retrieve  potential queries based on a threshold, and then perform separation accordingly.
>
> **Q5: Regarding the use of ImageBind?**
>
> **A5:** We use the pretrained ImageBind model. This is explicitly highlighted again at line 193 of the paper to emphasize our reliance on the ImageBind repository.
>
> We hope these responses address your concerns clearly! Please feel free to reach out with any additional questions.

---

> > ### Comment · Reviewer_7ccM · 2024-11-27
> >
> > Thank you to the authors for their thoughtful responses and the effort put into the rebuttal. Based on the rebuttal and the points raised by other reviewers, I believe the paper addresses most of the concerns effectively and makes a valuable contribution to this field of research. Therefore, I would like to maintain my recommendation towards accepting the paper.
> >
> > However, one of my initial points regarding the weighting factor for negative queries was not fully addressed in the rebuttal. I would appreciate further clarification from the authors regarding their insights into Figure 2. On top of that, I'm curious if there is an "optimal" weighting factor for certain types of queries? Could this value be deterministically derived during the inference phase to simplify the use of negative queries, rather than relying on trial-and-error approaches? Providing such insights could significantly enhance the usability and practical implementation of the proposed method.

---

> > > ### Author Response · Authors · 2024-12-01
> > > **Further Response to Reviewer 7ccM**
> > >
> > > Thank you once again for your thoughtful response. We are delighted to hear that most of your concerns have been resolved, and we deeply appreciate your recognition of our work. Allow us to provide further clarification on your remaining questions:
> > >
> > > **Q1-1: The difference between naive subtraction (**$Q{\prime} = Q - \alpha Q_N$**) and ours (**$Q{\prime} = (1 + \alpha) Q - \alpha Q_N$**)**
> > >
> > > **A1-1:** While the mathematical difference between these two formulations may seem minimal, their impact on performance is significant. As highlighted in lines 240–243, our formulation ($Q{\prime} = (1 + \alpha) Q - \alpha Q_N$) was carefully designed to address issues inherent to naive subtraction.
> > >
> > > Specifically, as shown in Figure 3, ImageBind embeddings are projected into a unified space during training, where all embeddings are mapped to a unit vector space $e$. When $\alpha$ > 1, naive subtraction ($Q{\prime} = Q - \alpha Q_N$) shifts the query embedding outside this unified vector space (resulting in the embedding space of $(1 - \alpha)e$), causing performance degradation and instability.
> > >
> > > In contrast, our formulation ($Q{\prime} = (1 + \alpha) Q - \alpha Q_N$) ensures that the modified query embedding remains in the same vector space as the original query. This alignment prevents issues caused by spatial mismatch and leads to significantly improved stability and performance.
> > >
> > > **Q1-2: Determining the “optimal” weighting factor for specific queries**
> > >
> > > **A1-2:** Thank you for raising this important question. We have indeed considered how to reduce the “manual effort” required during inference to determine the optimal weighting factor $\alpha$.
> > >
> > > As discussed in lines 406–417, naive subtraction ($Q{\prime} = Q - \alpha Q_N$) poses significant challenges in identifying the best $\alpha$, due to the lack of robustness to variations in this parameter. In contrast, our approach demonstrates strong robustness to $\alpha$, as performance remains stable for $\alpha$ > 0.5, minimizing the need for extensive parameter tuning during inference.
> > >
> > > From the performance curve in Table 2, it is evident that when $\alpha = 0.5$, it can be regarded as the optimal choice, delivering relatively strong performance and demonstrating robust consistency across different samples.
> > >
> > > Thank you again for your valuable feedback and questions. We hope this response fully addresses your concerns and further enhances your confidence in our work. Please feel free to reach out if you have any additional comments or questions.

---

### Public Comment · ~Xubo_Liu1 · 2024-11-27

Dear Authors,

The concept of “OmniSep” has been on my mind since developing the LASS-Net and AudioSep model series. Thank you for your contributions to multi-modal audio source separation and for making it works well.

After reading your paper, I have a few questions/concerns, particularly related to the experimental setup and the open-vocabulary separation claim:

1. **Pre-Trained AudioSep Comparison**: I’d be interested to see the performance of pre-trained AudioSep compared directly with the baseline and OmniSep, rather than reproducing AudioSep using CLIPSep's dataset partitioning. As the key contribution of AudioSep is scaling training to a large dataset and generalizing well in open-domain sound separation with language queries.

2. **Diverse Evaluation Sets**: To strengthen your claim in L53-56, I suggest conducting experiments on more diverse evaluation sets. AudioSep provides a dedicated test benchmark (https://drive.google.com/drive/folders/1PbCsuvdrzwAZZ_fwIzF0PeVGZkTk0-kL) that may be useful here. In my opinion, datasets like VGGSound and MUSIC are relatively small, and training on such datasets and evaluating on the same domain can often lead to better performance metrics - which may outperform models pre-trained on general datasets, but lacks the ability to generalize effectively across domains. For example, In my experience, fine-tuning AudioSep on the MUSIC dataset enabled me to achieve an SDRi of 18 dB, whereas pretraining results were around 9 dB.
3. **Effectiveness on DCASE 2024 Task 9 Datasets**: In DCASE 2024 Task 9, we created several synthetic and real datasets for evaluation with open-domain text queries.  I believe these datasets provide a good test set for OmniSep, and demonstrating OmniSep’s effectiveness on these datasets compared to AudioSep and other baselines could further substantiate your claims on open-vocabulary sound separation. DCASE 2024 Task 9 Evaluation set: https://zenodo.org/records/11425256
4. **Correction on Table 9**: In Table 9, it is suggested that AudioSep is unable to perform AQSS or IQSS. However, since AudioSep is trained with CLIP/CLAP text encoders (AudioSep-CLIP one has not been released), it can perform AQSS/IQSS, though its performance may not be as robust as when using text queries.

I appreciate your time and consideration in addressing these questions and look forward to your insights.

Best regards,

Xubo Liu

---

> ### Author Response · Authors · 2024-12-01
> **Response to Public Comment of Xubo (1/N)**
>
> Thank you for your interest in our work. AudioSep, WavCaps, and similar projects have introduced a new wave of progress in sound separation by expanding datasets to effectively achieve open-vocabulary sound separation. We sincerely respect and appreciate your contributions. Below, allow us to address your questions/concerns in detail:
>
> **Q1: Comparison with AudioSep**
>
> **A1**: Thank you for your inquiry. Please understand that we did not directly compare OmniSep’s performance with AudioSep for the following reasons:
>
> 1. Differences in training configurations:
>
>     AudioSep processes 32kHz data during training, with a signal-to-noise ratio (SNR) range of -10 to 10 dB, resulting in clean and controlled data, consistent with its proposed benchmark settings (the snr in test set is 0dB). In contrast, OmniSep follows the CLIPSep configuration, processing 16kHz data without imposing additional SNR range restrictions on the audio. When testing AudioSep’s official model on the VGGSOUND-clean test set, we observed suboptimal performance because VGGSOUND-clean includes samples that fall outside the SNR range used during AudioSep’s training. Consequently, we retrained AudioSep on the VGGSOUND dataset to ensure fair evaluation. **Apparently, there exists a degree of domain shift between the training and test distributions for OmniSep and AudioSep, making it challenging to compare the two models fairly on a unified test set.**
>
>     *Table R1. Comparison on VGGSOUND-clean（TQSS）*
>
>     | Method | SDR | SDRi | SI-SDR |
>     | --- | --- | --- | --- |
>     | AudioSep(Official Version) | 3.04 | 3.54 | 4.49 |
>     | OmniSep | **6.70** | **6.64** | **5.53** |
> 2. Discrepancy in training data size:
>
>     As you noted, AudioSep’s key contribution lies in scaling up to a large dataset (14,000 hours). OmniSep, in contrast, was trained solely on VGGSOUND (550 hours). This significant difference in training data size naturally impacts both separation performance and open-vocabulary capability. In the future, we plan to augment OmniSep’s training data using WavCaps. However, we refrained from including WavCaps data in this version to maintain fairness when comparing OmniSep with other IQSS baselines.
>
>
> To meet your expectations regarding performance comparisons, we have included results for AudioSep + Query-Aug in the next question (Q2). This demonstrates the effectiveness of our proposed method in enhancing open-vocabulary performance.

---

> ### Author Response · Authors · 2024-12-01
> **Response to Public Comment of Xubo (2/N, N=2)**
>
> **Q2: Comparison on diversified test sets**
>
> **A2**: Thank you for your suggestion. We would like to reiterate that **our primary contribution lies in enabling composed query sound separation across multiple modalities**. Query-Aug is a training-free method specifically designed to tackle the open-vocabulary challenges that arise due to the limited scale of omni-modality datasets.
>
> - To address your request, we integrated the Query-Aug method with AudioSep and evaluated its performance on the datasets you proposed. The results are presented in Table R2. Please note that since AudioSet and VGGSOUND use class labels as queries, these are in-domain queries already utilized during training. When applying Query-Aug for enhancement, the retrieved  $\text{query}_{\text{aug}}$  is the original query itself. As a result, the performance of AudioSep+Query-Aug on these datasets is identical to that of AudioSep. Therefore, we did not include evaluations on AudioSet and VGGSOUND datasets.
>
>     *Table R2: Comparison on the benchmark of AudioSep.*
>
>     | Method | MUSIC(SDRi) | MUSIC(SI-SDR) | ESC-50(SDRi) | ESC-50(SI-SDR) | Clotho(SDRi) | Clotho(SI-SDR) | AudioCaps(SDRi) | AudioCaps(SI-SDR) |
>     | --- | --- | --- | --- | --- | --- | --- | --- | --- |
>     | AudioSep(Paper version) | 9.75 | 8.45 | 10.24 | 9.16 | 6.51 | 4.84 | 7.68 | 6.45 |
>     | AudioSep(Github version) | 10.508 | 9.425 | 10.040 | 8.810 | 6.850 | 5.242 | 8.220 | 7.189 |
>     | AudioSep+Query-Aug(ours) | **10.719** | **9.712** | **10.720** | **9.840** | **7.201** | **5.795** | **8.221** | **7.190** |
>
>     Using query-aug, we observed significant performance improvements on MUSIC, ESC-50, and Clotho, highlighting its effectiveness in enhancing open-vocabulary capability. Interestingly, the performance gain on AudioCaps was minimal. We guess this is because for extremely fine-grained queries, the retrieved in-domain queries might fail to fully capture the original query’s semantics. (*We would also like to draw your attention to a potential issue in your benchmark: the AudioCaps test set appears to be included in the AudioSet training set. This overlap may explain the high performance of the original model, as these audio samples might have been exposed to AudioSep during training.*)
>
> - Additionally, we conducted specific tests to further illustrate the effectiveness of Query-Aug for addressing the open-vocabulary problem. By varying the queries for the same audio, we observed that even minor alterations, such as changing the case (from “Cello” to “cello”), could significantly affect model performance. Query-Aug mitigates this issue by retrieving in-domain queries, which are more robust in performance, thereby improving query comprehension and enabling more robust sound separation.
>
>     *Table R3: Comparison on varying queries for the same audio.*
>
>     | Query | Query-Type | SDR | SDRi | SI-SDR |
>     | --- | --- | --- | --- | --- |
>     | cello sound | Out-of-domain | 5.570 | 5.458 | 2.279 |
>     | a sound of cello | Out-of-domain | 6.213 | 6.100 | 2.279 |
>     | the sound of cello | Out-of-domain | 6.779 | 6.667 | 5.495 |
>     | cello | Out-of-domain | 7.113 | 7.001 | 6.097 |
>     | A cello is playing. | In Domain | 7.234 | 7.122 | 5.753 |
>     | Cellos are playing. | In Domain | 7.765 | 7.653 | 6.233 |
>     | Cello | In Domain | 7.958 | 7.845 | 7.026 |
>
> **Q3: Effectiveness on DCASE 2024 Task 9 Datasets**
>
> **A3:** Unfortunately, since this competition has concluded, we are unable to submit results for evaluation. Moreover, we were unable to access the ground truth audio for the evaluation set, which prevents us from calculating quantitative metrics. Despite our efforts, we regret that we could not locate the relevant resources.
>
> If such resources are publicly available, we would greatly appreciate your guidance in accessing them. Although we are currently unable to provide a performance comparison on this test set, we hope you agree that the results presented in Q2 sufficiently demonstrate the contributions and effectiveness of our method to open-vocabulary sound separation.
>
> **Q4: Results in Table 9**
>
> **A4:** We apologize for any confusion caused by Table 9. The “modality” mentioned refers to the query modality during inference and does not imply that AudioSep is limited to TQSS. In our paper, AudioSep was used as a strong baseline for TQSS. If needed, we are happy to clarify this with annotations in the camera-ready version.
>
> Thank you again for your detailed feedback and interest in our work. We hope our responses address your concerns and further clarify the contributions of our proposed methods. Please do not hesitate to reach out with any additional questions or comments.

---

### Meta-Review · Area_Chair_uiHX · 2024-12-19

**Metareview:**

This paper presents a novel system for query-based sound separation, called OmniSep, that accommodates multiple query modalities (text, image, and audio), either independently or in combination. Technical contributions consist in Query-Mixup for simultaneous multi-modal query processing, which is also supported by a negative query mechanism for sound suppression at inference, and Query-Aug for handling natural language descriptions beyond naive class labels, so enabling open-vocabulary sound separation.
Experimental validation on several datasets demonstrates OmniSep's iincreased performance as compared to existing methods (single-modality) across all query modalities.

This work is appreciated under several respects, including the interesting addressed task, the technically sound and novel methodological contributions, and the experimental analysis provided.

Weak aspects regard, in general, several requests of better addressing some specific parts of the methodology (for all three main contributions, but mainly for Query-Aug), and clarifications/revision of the experimental section.

The authors replied properly to all such comments and the initial scores (6, 5, 5, 5) became all positive (6, 6, 6, 6).

Therefore, this paper can be considered acceptable for publication to ICLR 2025.

**Additional Comments On Reviewer Discussion:**

See above

---

### Decision · Program_Chairs · 2025-01-22

Accept (Poster)